# Contrasting responses of woody and herbaceous vegetation to altered rainfall characteristics in the Sahel

Wim Verbruggen[1,2], Guy Schurgers[2], Stéphanie Horion[2], Jonas Ardö[3], Paulo N. Bernardino[4,5], Bernard Cappelaere[6], Jérôme Demarty[6], Rasmus Fensholt[2], Laurent Kergoat[7], Thomas Sibret[1], Torbern Tagesson[2,3], and Hans Verbeeck[1]

[1]CAVElab, Department of Environment, Ghent University, Ghent, 9000, Belgium
[2]Department of Geosciences and Natural Resource Management, University of Copenhagen, Copenhagen, 1350, Denmark
[3]Department of Physical Geography and Ecosystem Science, Lund University, Lund, 22100, Sweden
[4]Department of Earth and Environmental Sciences, KULeuven, Leuven, 3000, Belgium
[5]Laboratory of Geo-Information Science and Remote Sensing, Wageningen University, Wageningen, 6708, The Netherlands
[6]HydroSciences Montpellier, IRD/CNRS, Université de Montpellier, Montpellier, 34090, France
[7]Géosciences Environnement Toulouse, CNRS/UPS/IRD, Toulouse, 31400, France

*Correspondence to*: Wim Verbruggen (wim.verbruggen@ugent.be)

**Abstract.** Dryland ecosystems form a major land cover, accounting for about 40% of Earth's terrestrial surface and net primary productivity, and housing more than 30% of the human population. These ecosystems are subject to climate extremes (e.g. large-scale droughts, extreme floods) that are projected to increase in frequency and severity under most future climate scenarios. In this modelling study we assessed the impact of single years of extreme (high or low) rainfall on dryland vegetation in the Sahel. The magnitude and legacy of these impacts were quantified on both the plant functional type and the ecosystem levels. In order to understand the impact of differences in the rainfall distribution over the year, these rainfall anomalies were driven by changing either rainfall intensity, event frequency or rainy season length. The Lund-Potsdam-Jena General Ecosystem Simulator (LPJ-GUESS) dynamic vegetation model was parameterized to represent dryland plant functional types (PFTs) and was validated against flux tower measurements across the Sahel. Different scenarios of extreme rainfall were derived from existing Sahel rainfall products and applied during a single year of the model simulation timeline. Herbaceous vegetation responded immediately to the different scenarios, while woody vegetation had a weaker and slower response, integrating precipitation changes over a longer timeframe. An increased season length had a larger impact than increased intensity or frequency, while impacts of decreased rainfall scenarios were strong and independent of the season characteristics. Soil control on surface water balance explains these contrasts between the scenarios. None of the applied disturbances caused a permanent vegetation shift in the simulations. Dryland ecosystems are known to play a dominant role in the trend and variability of the global terrestrial $CO_2$ sink. We showed that single extremely dry and wet years can have a strong impact on the productivity of drylands ecosystems, which typically lasts an order of magnitude longer than the duration of the disturbance. Therefore, this study sheds new light on potential drivers and mechanisms behind this variability.

## 1 Introduction

Dryland ecosystems account for about 40% of Earth's terrestrial surface and net primary productivity (Grace et al., 2006; Wang et al., 2012), and shelter more than 30% of the human population (Gilbert, 2011). These ecosystems are subject to climate extremes that are projected to increase in frequency and severity under most future climate scenarios (IPCC, 2014; Sillmann et al., 2013). Such extremes (e.g., large extent droughts, extreme floods) can have a devastating impact on the ecosystems and livelihoods of global drylands, as well as amplifying pressure on fragile economic structures (Ibrahim, 1988; United Nations Office for the Coordination of Humanitarian Affairs, 2013). The Sahel, situated south of the Sahara desert, is one of the largest dryland areas of the world, covering more than 3 million $km^2$. It is home to a population of around 135 million people, which is expected to increase by a factor of 2.3 between 2014 and 2050 (Haub and Toshiko, 2014).

The Sahel is mostly dominated by savanna grasslands. These complex biomes consist of a sparse cover of $C_3$ trees and shrubs, overlying an understory dominated by $C_4$ grasses. The co-existence of herbaceous and woody species in drylands has been the subject of many studies (e.g., Dodd et al., 1998; McMurtrie and Wolf, 1983; van Wijk and Rodriguez-Iturbe, 2002), and can be explained by different strategies in root-water access and phenology. Disturbances such as wildfires can have a

major impact on the tree cover as well, especially in mesic regions (mean annual precipitation (MAP) > 650 mm) (Sankaran et al., 2008). Capturing these complex ecosystems with dynamic vegetation models can be challenging, yet rewarding (Whitley et al., 2017) as these models can provide novel insights in the dynamics of tree-grass competition for resources, as

well as dryland ecosystem carbon and water cycling in general.

Although the vegetation structure and ecosystem productivity in water-limited ecosystems is mainly driven by annual total precipitation (Lehmann et al., 2014; Sankaran et al., 2005, 2008), intra-annual rainfall variability, which is characterized by the variability in rain event intensity, frequency and timing of the wet season, has a large impact on the vegetation as well, by changing the spatial and temporal availability of soil water for plant uptake (Berry and Kulmatiski, 2017; Case and

Staver, 2018; Guan et al., 2018; Kulmatiski and Beard, 2013; Xu et al., 2018; Zhang et al., 2018, 2019). The year-to-year variation in these characteristics is significant in global drylands, including the Sahel (Reynolds et al., 2007; Zhang et al., 2017). Climate projections for the end of the 21st century generally show a delay in timing of the rainy season, with average shifts of around 5 to 10 days for the Sahel (Dunning, 2018; IPCC, 2014; Pascale et al., 2016). Total precipitation is expected to decrease in the western parts and to increase in the central and eastern parts of the Sahel, although a high variability

remains among the different climate model predictions (Biasutti, 2019; Pascale et al., 2016). Furthermore, an increase in rain event intensity, coupled with a decrease in frequency has been observed in recent years (Panthou et al., 2014; Taylor et al., 2017) and is projected for the coming century (Dunning, 2018). Even though the region has a long history of adapting to drastic changes in rainfall (Mortimore, 2010), it is still uncertain how current and future changes in rainfall regimes will impact the plant functional responses in the Sahel and in drylands in general.

Dryland vegetation is known to respond in contrasting ways to intra-annual rainfall variability. An increased frequency of heavy rainfall events is reported to facilitate woody encroachment in savannah ecosystems (Kulmatiski and Beard, 2013; Zhang et al., 2019), but this response is modulated by the underlying soil texture, as a more intense rainfall leads to a lower tree cover on soils with a finer texture (Case and Staver, 2018). Other studies found that regions with a given amount of total seasonal rainfall have a higher woody cover under a more frequent but less intense rainfall climatology, which can be

explained by differentiated tree and grass water use strategies (Good and Caylor, 2011; Xu et al., 2015, 2018). D'Onofrio et al. (2019) found a positive relationship between grass cover and rain event frequency, but only a weak link between tree cover and rainfall seasonality characteristics for drylands (MAP $\leqq$ 650 mm). Zhang et al. (2018) found vegetation in drylands to be impacted significantly by the number of rainy days and timing of the wet season. A vegetation model study by Guan et al. (2018) did not find a significant difference in impacts between the different seasonal rainfall characteristics

mentioned above, although mesic regions depicted a stronger increase in gross primary productivity (GPP) with enhanced length of the season than with enhanced rain event intensity or frequency.

In order to gain a more detailed process-based insight in how dryland vegetation is affected by the distribution of rainfall over the rainy season, we used a dynamic vegetation model to study the impact and legacy of single anomalous rainy seasons on the vegetation. The approach presented here is therefore complementary to earlier studies, such as Guan et al. (2018),

which mainly assessed the impact on the vegetation of long-term changes in intra-seasonal rainfall variability, informing on

the ecosystem state under prolonged changes in rainfall regime. Hence, the vegetation response in such studies is subject to cumulative effects of repeated rainfall disturbances, obscuring the underlying mechanisms that drive these responses.

We aimed at assessing the impact of different rainfall scenarios on the vegetation response at four flux tower sites across the Sahel (Tagesson et al., 2016; Table 1), investigating the response of individual plant functional types (PFTs), and of the ecosystem as a whole. We parameterized the LPJ-GUESS dynamic global vegetation model (Smith et al., 2014) for the Dahra site in Senegal (Tagesson et al., 2015), using field measurements and a literature study. The model was evaluated at all Sahel sites by testing whether it significantly improved the representation of the site ecosystem fluxes relative to the published version of the model (Smith et al., 2014). The model experiments were set up as a disturbance event, where we altered the rainfall during one year in the meteorological driver time series. We changed the total rainfall together with one of the underlying seasonal characteristics (i.e. intensity, frequency or length), while keeping the other two characteristics invariant.

Adopting this approach, we addressed the following research questions: (1) how do years of extreme rainfall with different seasonal characteristics impact the fluxes and composition of dryland ecosystems in the Sahel in the period following the extreme event, and (2) how do the magnitude and legacy of these impacts vary across the different plant functional types?

**Table 1.** Overview of the different flux tower sites used in this study, together with the 1979-2016 mean annual precipitation and its standard deviation (MAP, mm year$^{-1}$) from the MSWEP v1.2 dataset (Beck et al., 2017), mean annual temperature (MAT, °C) from the WFDEI dataset (Weedon et al., 2014), FAO soil classification (FAO, 1988), ecosystem type and tree cover (Tagesson et al., 2016), rainfall seasonality (fraction of rainfall inside the rainy season), measurement years (eddy covariance data available for model validation), and literature reference.

| Site | Dahra, Senegal | Agoufou, Mali | Wankama, Niger | Demokeya, Sudan |
|---|---|---|---|---|
| **Coordinates** | 15.40°N 15.43°W | 15.34°N 1.48°W | 13.65°N 2.63°E | 13.28°N 30.48°E |
| **MAP ± 1σ (mm/y)** | 339 ± 107 | 258 ± 83.4 | 303 ± 67.8 | 164 ± 65.1 |
| **MAT (°C)** | 28.7 | 30.2 | 29.5 | 28.1 |
| **Soil classification** | Luvic Arenosol | Ferralic Arenosol | Ferralic Arenosol | Cambic Arenosol |
| **Ecosystem type** | Shrubland savanna | Open woody savanna | Fallow bush | Sparse acacia savanna |
| **Tree cover** | 3% | 4% | <1% | 7% |
| **Seasonality** | 0.94 | 0.93 | 0.93 | 0.92 |
| **Measurement years** | 2010 - 2013 | 2007 | 2005 - 2012 | 2007 - 2009 |
| **References** | Tagesson et al., 2015 | Mougin et al., 2009 | Boulain et al., 2009 Ramier et al., 2009 | Sjöström et al., 2009 |

## 2 Methods

### 2.1 Study area

The Sahel is a semi-arid ecoclimatic transition zone, bridging the Sahara desert in the north with the Sudanian savanna in the
south. It is usually defined by the 150 mm and 700 mm isohyets delineating its northern and southern borders, respectively.
In this study we used data from four flux tower sites that have been established in the Sahel, measuring land-atmosphere
carbon, water and energy exchanges, together with meteorological data (Tagesson et al., 2016; Table 1). The flux towers are
located at Dahra in Senegal (DAH), Agoufou in Mali (AGG), Wankama in Niger (WFF), and Demokeya in Sudan (DEM)
(Fig. 1). All sites consist of a grassy savanna with a sparse tree cover, growing on sandy arenosol soils. Annual total rainfall
varies from 339 mm in the west (Dahra) to 164 mm in the east (Demokeya), with mean annual temperatures around 29°C
(Table 1).

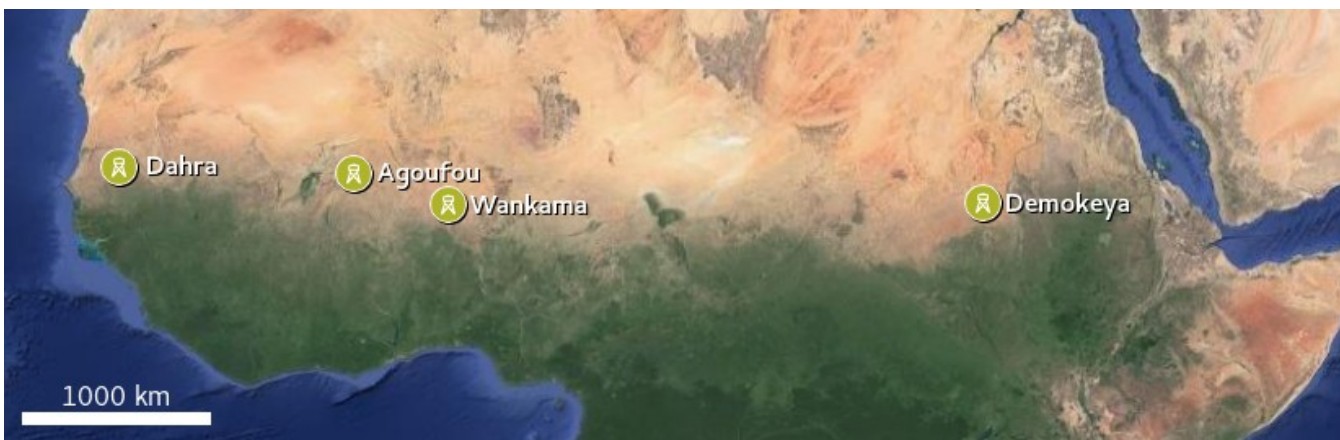

**Figure 1.** Map including the Sahel region, showing the locations of the different flux tower sites: Dahra (Senegal), Agoufou
(Mali), Wankama (Niger), and Demokeya (Sudan). Imagery ©2020 TerraMetrics, Map data ©2020 Google.

### 2.2 Vegetation model

We used the LPJ-GUESS process-based dynamic global vegetation model, which simulates the global vegetation structure
with its associated carbon, nitrogen and water cycles (Smith et al., 2014). Similar to many global models, LPJ-GUESS uses
plant functional types to represent physiological, morphological and phenological differences in vegetation. Out of the 12
standard PFTs in LPJ-GUESS, three are relevant for the Sahel: tropical broadleaved evergreen trees, tropical broadleaved
deciduous trees, and C4 grasses. Within each gridcell, LPJ-GUESS simulates plant growth based on competition for light,
space, water and soil nitrogen between individuals from different PFTs. Processes related to photosynthesis, soil hydrology,

respiration, stomatal conductance and phenology are simulated on a daily time step, while carbon allocation, establishment, mortality and wildfire disturbance (Thonicke et al., 2001) are accounted for at the end of each simulated year. To account for the heterogeneity in age distribution of ecosystems, for each run 100 replicate patches were forced with the same meteorological data but exposed to stochastic differences in disturbances.

Phenology of the drought-deciduous PFTs is based on a water stress scalar in the model. Low values of this scalar represent stress due to reduced soil water content, leading to a reduction of photosynthesis through stomatal closure. When this variable drops below a given threshold, the dry season starts and deciduous trees will shed their leaves. Likewise, when this scalar rises above this threshold new leaves will be produced, taking into account a prescribed minimum dormancy period (Smith et al., 2014).

Soil hydrology is represented by a two-layer bucket model with percolation between the layers and drainage at the bottom (Gerten et al., 2004). The upper layer has a depth of 0.5 m, while the lower layer is 1 m deep, adding up to a total soil depth of 1.5 m. Rainfall will replenish plant-available water in the upper layer up to field capacity, above which excess water will be expelled as surface runoff. The lower soil layer is supplied with water by percolation from the upper layer. Transpiration by plant canopies will in turn reduce the water content in both soil layers. Different PFTs can have different root biomass distributions across the soil layers, e.g. grasses will have 90% of their root biomass in the upper layer, while trees have deeper roots in the model (Table 2). LPJ-GUESS has previously been used in Sub-Saharan Africa and other savanna studies and the model is known to give a reasonable representation of large-scale sensitivities to drought in drylands at the global scale, and for Africa specifically (Ahlstrom et al., 2015; Baudena et al., 2015; Boke-Olén et al., 2018; Brandt et al., 2017, 2018; Lehsten et al., 2016). Nonetheless, the parameterization of the PFTs has never been optimized for the drylands in the Sahel specifically.

**Table 2.** Important PFT parameter values used in LPJ-GUESS: photosynthetic pathway, specific leaf area (SLA, $m^2$ $kgC^{-1}$), wood density (WD, $kgC$ $m^{-3}$), maximum daily evapotranspiration rate (emax, mm $day^{-1}$) and root distribution (RD, fraction of the root biomass in the upper and lower soil layer, respectively) (Gerten et al., 2004; Nielsen, 2016; Sibret, 2017; Sibret et al., 2020).

| PFT | Photo | SLA | WD | emax | RD |
|---|---|---|---|---|---|
| $C_4$ grass | $C_4$ | 35.3 | - | 7 | 0.9 : 0.1 |
| Evergreen Trees | $C_3$ | 13.9 | 319.1 | 5 | 0.6 : 0.4 |
| Deciduous Trees | $C_3$ | 25.7 | 318.7 | 5 | 0.6 : 0.4 |

**2.3 Model parameterization and validation**

We adjusted the parameterization of LPJ-GUESS to the local conditions by updating two plant functional traits (specific leaf area and wood density) to values from Nielsen (2016) and Sibret (2020). The tropical evergreen tree PFT was based on

*Balanites aegyptiaca*, while the deciduous tree PFT was based on *Acacia tortilis* and *Acacia senegal*, which are the main woody species found at Dahra, Senegal. For the C4 grass parameters we used the average trait values of all C4 grasses identified by Sibret (2020), and the maximum daily evapotranspiration rate from Gerten et al. (2004). The most important parameters to differentiate the PFTs are given in Table 2. Similar species, or at least a functionally similar vegetation composition, can be found at the other three sites. As all sites are characterized by sandy arenosols (Table 1) and the used soil database does not take into account differences in lower-level soil classification, all sites were assumed to have the same sandy soil texture in the simulations (90% sand, 5% silt, 5% clay).

The model was evaluated against flux tower data from the four Sahel sites by comparing a 10-day moving average of the measured daily net ecosystem productivity (NEP) and evapotranspiration (ET) time series with model predictions. Model performance metrics were summarized in a Taylor diagram (Taylor, 2001).

## 2.4 Model forcing timeline

By default, LPJ-GUESS is driven by daily interpolations of monthly Climatic Research Unit and National Centers for Environmental Prediction (CRU-NCEP) meteorological forcing data (Viovy, 2018). To improve the temporal resolution of the meteorological forcing, we used meteorological data extracted from WATCH Forcing Data methodology applied to ERA-Interim reanalysis (WFDEI; Weedon et al., 2014) with substituted Multi Source Weighted Ensemble Precipitation v1.2 data (MSWEP; Beck et al., 2017, 2019). Both reanalysis datasets contain daily averages of meteorological data from 1979 to 2016 (Fig. 2). The data have a 0.5° by 0.5° spatial resolution and only the grid cells containing the flux tower locations were selected. Vegetation demography and its associated carbon, water and nitrogen stocks were initialized in the model by a 500-year spin-up run from bare soil. During this spin-up run the meteorological forcing data were used in a cycle, combined with a constant $CO_2$ concentration of 296 ppm, corresponding to the 1901 level. The spin-up run was followed by a historical run from 1901 until 2016, using the same cycled meteorological forcing, but following the historical $CO_2$ record. After this historical run, the rainfall perturbation experiments were implemented. In order to account for variability in meteorological conditions prior to the perturbation, all years with near-average rainfall (MAP ± 1σ) of the original meteorological driver cycle were perturbed, each leading to a different ensemble member in the simulation. For this period, a constant $CO_2$ concentration (the 2016 value of 404 ppm) was applied (Fig. 3).

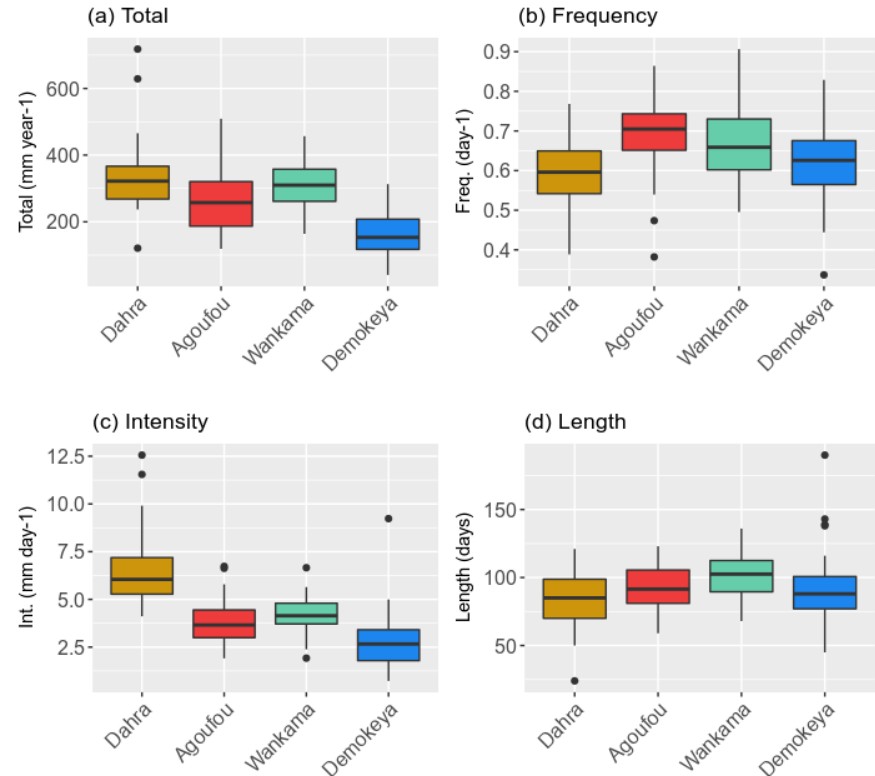

**Figure 2.** Median and variability of the rainy season characteristics for the Sahel sites (Table 1) studied: (a) total annual rainfall, (b) event frequency, (c) event intensity and (d) rainy season length (Table 3). Hinges represent the first and third quartiles, whiskers represent the largest (smallest) value at most 1.5 times the interquartile range above (below) the hinges, and dots represent outliers. Based on the daily MSWEP rainfall data for the period 1979-2016.

180

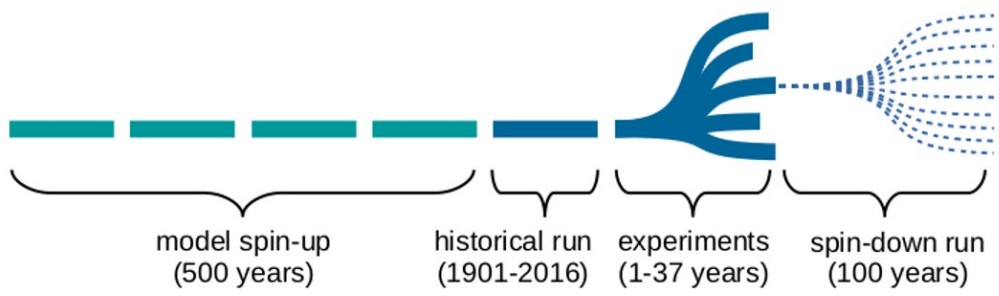

**Figure 3.** Overview of the general simulation timeline for each scenario. During the experiment period, one single year of the meteorological time series is disturbed, as illustrated by the different branches, and immediately followed by an

ensemble of spin-down runs, consisting of average rainfall years only. Each horizontal segment represents a cycle of meteorological forcing data.

## 2.5 Disturbance experiment set-up

The rainfall disturbance experiments were developed to depict an increase or decrease of the total precipitation in a given year by two standard-deviations of the annual rainfall, therefore representative of extreme years in the historical time series. This disturbance was applied in such a way that only one of the three seasonal characteristics (intensity, frequency or length; Table 3) changed, while the other two remained invariant, thus creating a target rainy season for the selected year (Table 4). A detailed description of the used algorithms can be found in Supplementary Materials S1. In order to preserve the internal meteorological consistency with the other drivers (air temperature and incoming short-wave radiation), we resampled all data from the original meteorological drivers: for each DOY in the goal scenario, we found a date with matching rainfall (±1 mm) in the original dataset and used all its meteorological data. Some restrictions were introduced in order to preserve general synoptic patterns: the resampled day should be close to the original DOY (±3 days, ±1 year) and if none were found, the neighbouring pixels from the reanalysis dataset were consulted. If still no day with matching rainfall was found, the time interval was gradually extended in days until a match was found, up to a maximum of ±40 days.

**Table 3.** Definitions of the different rainy season characteristics used in this study.

| Characteristic | Definition |
| --- | --- |
| Total rain | Total rainfall within the rainy season (mm). |
| Season start | Day of year after the minimum in climatological anomalous accumulation. Similar definition for the season end (DOY). |
| Season length | Difference between season start and end day of the year (days). |
| Intensity | Average daily total rainfall over all rainy days within the rainy season (mm day$^{-1}$). |
| Frequency | Inverse of average time (days) between rain events within the rainy season (day$^{-1}$). |

The six simulation scenarios (Table 4) were applied to each year of the meteorological cycle that had a total rainfall close (±1σ) to the time series average, in order to avoid applying additional perturbations to already extreme years. Depending on the variability in annual rainfall for each site, this results in Ne≈30 simulations (Fig. 3). Each disturbed year was immediately followed by a set (Ns=10) of spin-down runs, each run consisting of a random 100-year sequence with rainfall within one standard deviation of the time series average. This was implemented to average out any effects of the post-disturbance years' rainfall characteristics on our impact study. For this 100-year sequence, the same constant $CO_2$ concentration (the 2016 value of 404 ppm) was applied.

**Table 4.** Overview of the scenarios with actual rainy season characteristic values for the Dahra site (average and standard deviation taken over all ensemble members).

| Scenario | Modified characteristic | Change in total rainfall |
|---|---|---|
| Len+ | +73.3 ± 18.3 days | +186 ± 25.6 mm |
| Freq+ | +0.365 ± 0.079 events/day | +203 ± 26.6 mm |
| Int+ | +3.321 ± 1.737 mm/event | +177 ± 57.9 mm |
| Len- | -62.5 ± 21.8 days | -206 ± 11.8 mm |
| Freq- | -0.426 ± 0.077 events/day | -209 ± 11.8 mm |
| Int- | -4.179 ± 0.974 mm/event | -209 ± 10.3 mm |


This leads to an internally consistent set of meteorological model drivers for all six disturbance scenarios (Table 4), each simulated by an ensemble of $N_e \times N_s \approx 300$ runs at four Sahel sites (Fig. 3). For each of these disturbance runs, a reference run was included, based on exactly the same meteorological data but without applying the perturbation in the disturbance year (Fig. 4).

For each site and each scenario, the impact of the disturbance and its legacy on vegetation were finally quantified as the difference between the output of the reference and disturbance runs, displayed as a function of time since the disturbance event and finally averaged over all ensemble members (Fig. 4). Impact legacy is calculated as the last year for which the average impact is larger than its standard deviation, i.e. when the uncertainty on the impact becomes larger than its difference with the reference run, resulting in a relatively conservative estimation of legacy. We analyzed the response of

individual PFTs, as well as the ecosystem as a whole, by quantifying the impact on leaf area index (LAI), carbon cycling and surface water balance. We show the full impact time-series for the Dahra site as an illustration, together with a summarized result to compare key response descriptors (maximum impact and legacy) between the different sites. Full impact time-series for all sites can be found in Supplementary Materials.

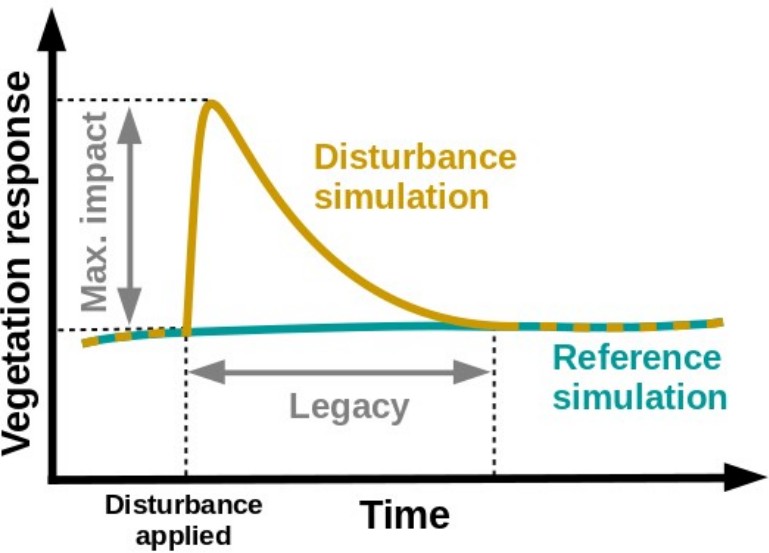

**Figure 4.** Concept of the model experimental setup, consisting of a disturbance simulation in which a rainfall disturbance is applied and a reference run, which is based on the same meteorological drivers but without any disturbance applied. Vegetation impact is described by maximum impact (amplitude) and legacy (years). The average impact is derived by subtracting the output of the reference simulations from the output of the disturbance simulations for each ensemble member, and then taking the average of the result over all ensemble members. The legacy is calculated as the last year for which the
average impact is larger than its standard deviation.

## 3 Results

### 3.1 Model evaluation

The updated parameterization of LPJ-GUESS captures the net ecosystem productivity (NEP) and evapotranspiration (ET) that were measured at the Dahra flux tower site, to which the model was parameterized (Fig. 5). Carbon uptake follows the
timing of the rainy season, but the amplitude of both NEP and ET are underestimated over the whole time series. The uncertainty on the Dahra flux tower NEP measurements varies around an average of 1.5 $gC/m^2/day$ at the peak of the rainy season, so the underestimation is significant (Tagesson et al., 2016b). However, the fluxes measured at the Dahra site are relatively high when compared to the other Sahel sites (Fig. S1; Tagesson et al., 2016a). When evaluated against daily NEP measurements of the other Sahel sites, this parameterization significantly improves the agreement between simulated and
observed daily carbon fluxes with respect to the published model parameterization, also when the latter is driven by the WFDEI-MSWEP meteorological forcing data which are used in this study (Fig. 6, Fig. S1). Interestingly, the improvement with respect to the published parameterization is larger for the other Sahel sites than for the Dahra site. In addition, we note that yearly averaged simulated values of surface runoff vary between sites from 16 mm/year to 49.8 mm/year, although the

runoff distributions have a long tail due to extreme years. Indeed, median runoff values are much lower, varying between 4.4

mm/year and 30 mm/year (Fig. S2).

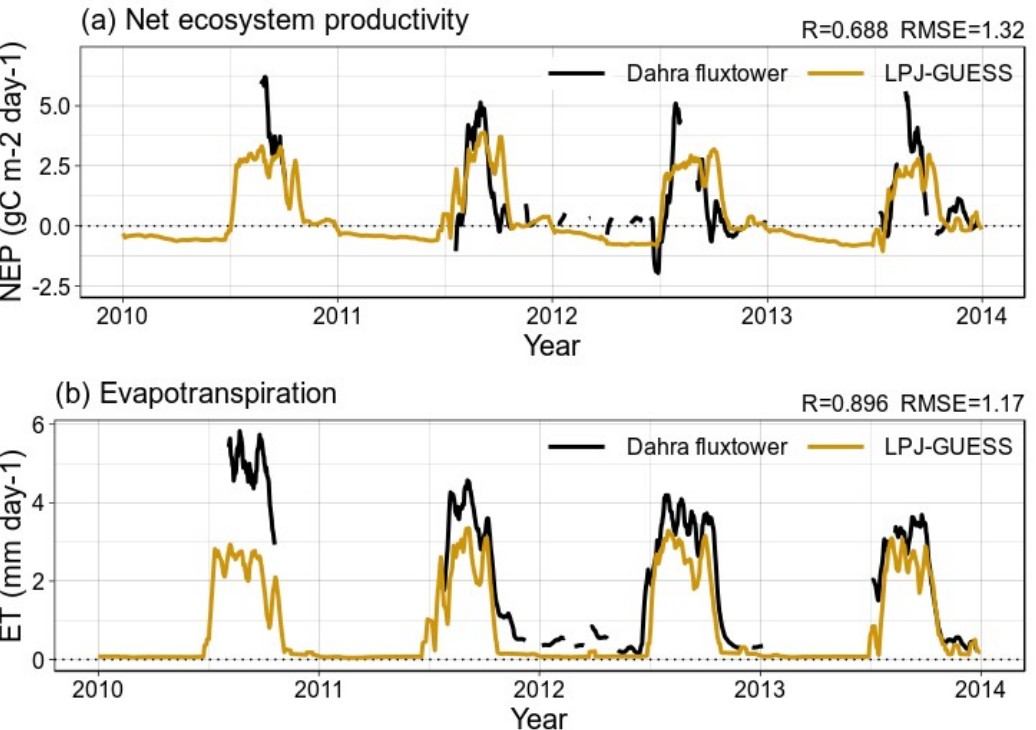

**Figure 5.** Time series of a 10-day moving average of (a) daily net ecosystem productivity (NEP) and (b) daily evapotranspiration (ET), comparing measurements from the flux tower near Dahra, Senegal with model results from LPJ-GUESS, using the Sahel-specific parameterization and WFDEI-MSWEP meteorological drivers.


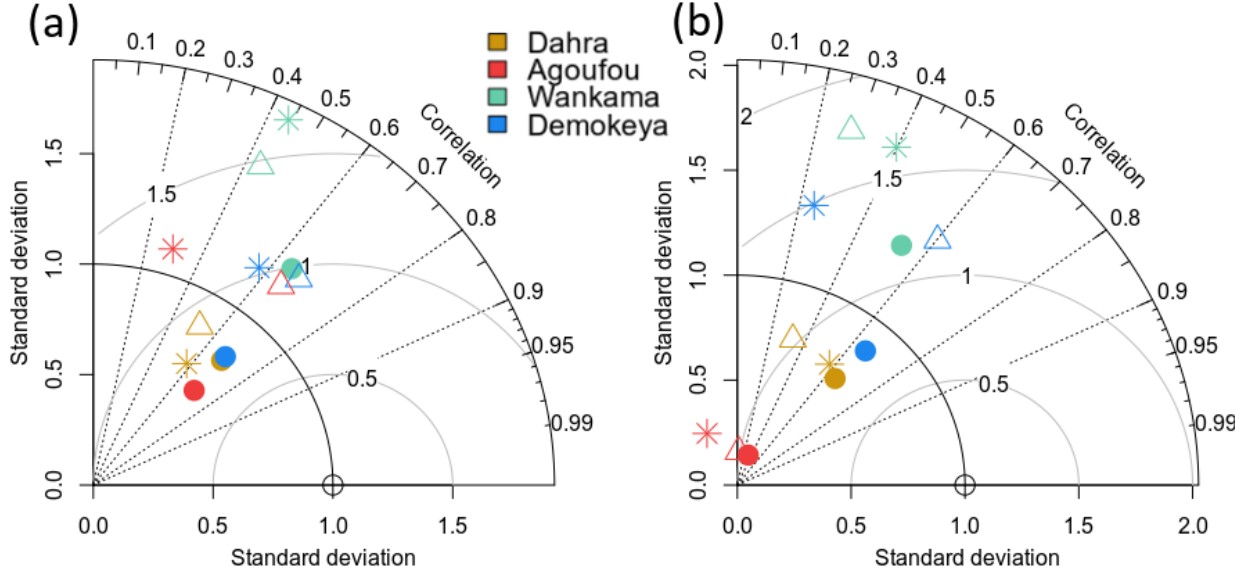

**Figure 6.** Taylor diagrams showing the correspondence between modelled and observed daily NEP values (10-day moving averages) for the Sahel flux tower sites, evaluated over (a) the whole year and (b) the rainy season only. We show a comparison between runs based on published model parameters using CRU-NCEP meteorological data (∗), published model version parameters using WFDEI-MSWEP drivers (△), and runs based on the new drylands-specific parameterization using the WFDEI-MSWEP drivers (●). Values were normalized so that the standard deviations of the observations equal unity, i.e. observations are located at the (1,0) coordinate on this figure Grey arcs represent the root mean square error (RMSE) between model output and observations.

## 3.2 Experiment results

From all applied scenarios, an increase in rainy season length caused the largest increase in LAI at all sites and for all PFTs. Especially $C_4$ grasses had a particularly strong response of ~120% for all sites, except for Wankama (Fig. 7, 8). This impact of the season length was the weakest at the Wankama site for all PFTs. For evergreen trees the amplitude of this impact varied strongly across the sites, from 30% at Wankama to 110% at Demokeya, while the legacy was of the same order as for the grasses (2-4 years). For deciduous trees this impact was less variable and generally lower in amplitude (10-30%), but longer in legacy, up to 14 years at the Dahra site (Fig. 8).

Scenarios of increased rain event frequency generally had a slightly larger impact than those of increased intensity. The impact of both scenarios was the weakest for deciduous trees (15-25%) and slightly stronger for evergreen trees (25-45%) and grasses (30-55%), with an exception for the Demokeya site, which showed a larger impact of these scenarios on the evergreen trees (65% for increased intensity and 85% for increased frequency) (Fig. 8).

In contrast, for the scenarios of decreased total rainfall, the simulated reduction in LAI was largely independent of the rainy season characteristic that was adjusted, although scenarios of decreased intensity depicted a slightly lower decline in LAI

overall (Fig. 7, 8). For C₄ grasses, both the maximum impact and the legacy of the impact varied between sites: Wankama experienced the least negative change in LAI (-45%) with a legacy of 3 years, while for Demokeya the amplitude and legacy were higher (-75% and 6 years, resp.; Fig. 8). For the tree PFTs the maximum impacts were of the same order for all sites and scenarios, with values around -50% for evergreen trees, and -30% for deciduous trees. On the other hand, the legacies of the negative disturbances on the trees varied across the sites, from 3 years at Demokeya to 7 years at Dahra, but were largely independent of how rainfall was reduced (except for a decreased rainfall intensity at the Demokeya site; Fig. 8).

For all scenarios and all sites, grasses responded immediately to changes in precipitation, with the highest impact occurring during the perturbed year (Fig. 7d, S5-S7). In contrast, the tree PFTs exhibited their peak impact in the year following the disturbance (Fig. 7e, 7f, S5-S7). At all sites, deciduous trees experienced a reversal in response ("overshoot") following the initial impact, which is up to the same order of magnitude as the initial impact, in particular for scenarios of increased rainy season length and rainfall reduction (Fig 7f, S5-S7). Depending on the site and scenario, these overshoots can last for multiple (>5) years, although they remain within model variability limits.

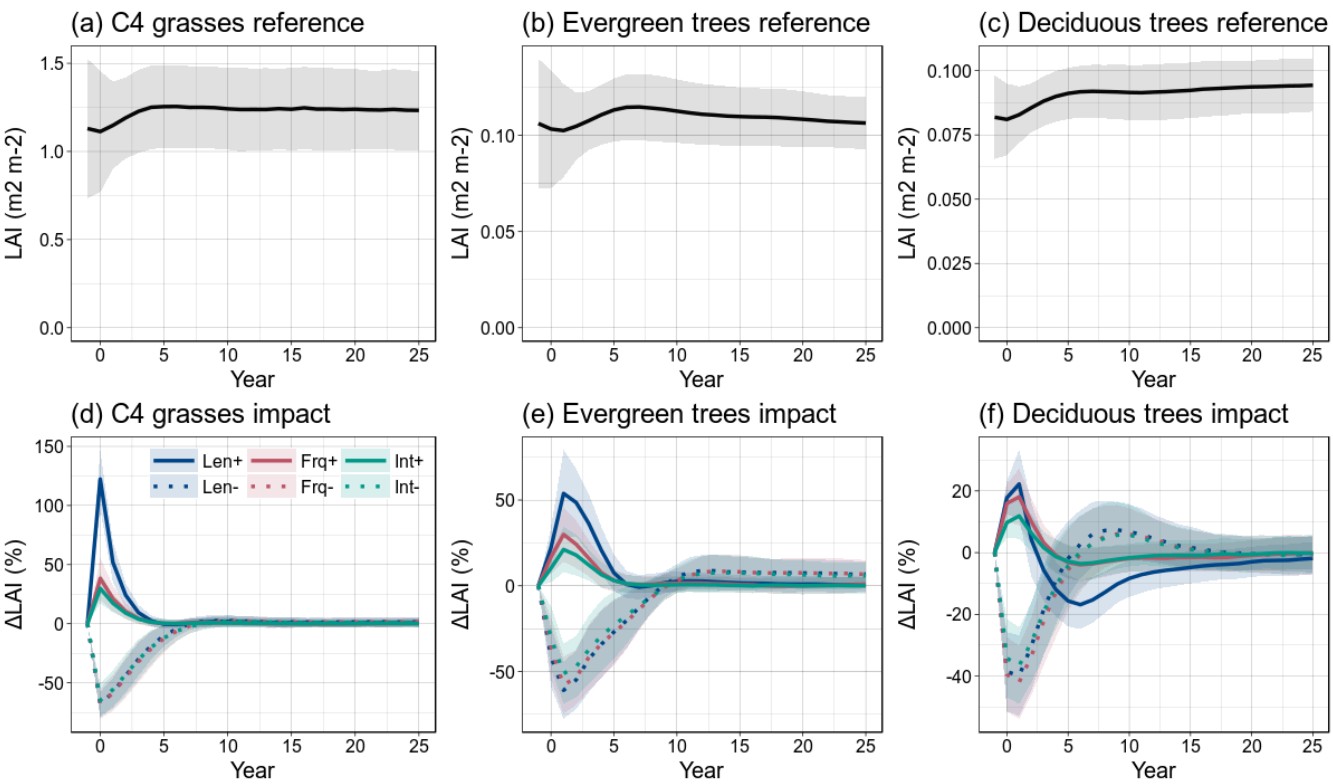

**Figure 7.** Response of the vegetation to the different rainfall scenarios for the Dahra site, in function of years since the disturbance event. (a-c) reference LAI of each PFT, averaged over all ensemble members; (d-f) vegetation response as the

mean relative LAI difference between the scenario runs and the reference runs. Shaded areas indicate variability of the model runs over all ensemble members (±1σ).

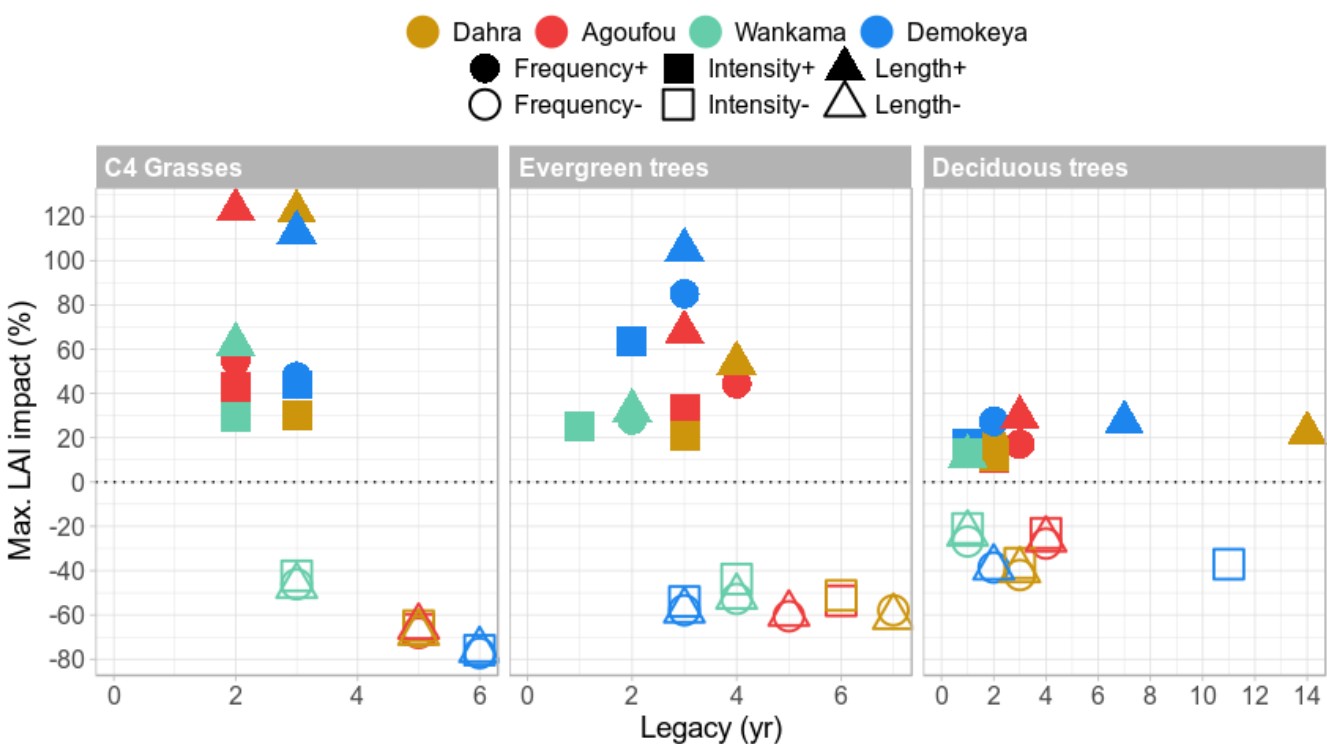

**Figure 8.** Summarized overview of the vegetation response to the different rainfall scenarios for all sites, showing the maximum impact (%) on the LAI for each PFT, along with its legacy (years), which is defined as the last year for which the impact is one standard deviation away from the reference value. Sites are represented by different colours, scenarios are represented by symbols.

On the ecosystem level, the reference net ecosystem productivity (NEP) was mostly positive but relatively small, ranging between -0.008 and 0.026 kgC/m$^2$/year for all sites (Fig. 9a, S8-S10), while reference yearly photosynthesis (gross primary productivity, GPP) varied between 0.16 and 0.47 kgC/m$^2$/year. Rainfall disturbance scenarios had a major impact on ecosystem productivity, with maximum NEP impact values typically an order of magnitude larger than the reference values (Fig. 9b, 10), ranging from 10% to 47% of the typical reference GPP. Increased season length had again the largest impact on NEP, and this contrast between the positive scenarios was most pronounced at the Dahra site. For the reduced rainfall

scenarios, the contrast in impact varied mainly between the sites, and again mostly by differences in legacy rather than amplitude. For scenarios of increased rainfall it takes 4 to 7 years to converge back to reference NEP values, while for

scenarios of reduced rainfall the impact legacy varies from 2 years at the drier sites (Demokeya, Agoufou) up to 10 years at the wetter sites (Dahra, Wankama). For all sites and all disturbance scenarios, the NEP also displayed a substantial reversal

("overshoot") in impact response, starting 2-3 years after the disturbance (Fig. 9b, S8-S10). The maximum amplitude of this reversal varies between 11% and 51% of the original impact and scales linearly with the initial impact value, i.e. scenarios of increased season length will have the strongest overshoot, while scenarios of decreased rainfall will have overshoots that differ between the sites, but not between the disturbance scenarios for each site (Fig. S8-S10). The net effect of this reversal on the cumulative NEP is to balance out the initial impact on the long run, which can take several decades, depending on the

site and scenario (Fig. 9d, 10).

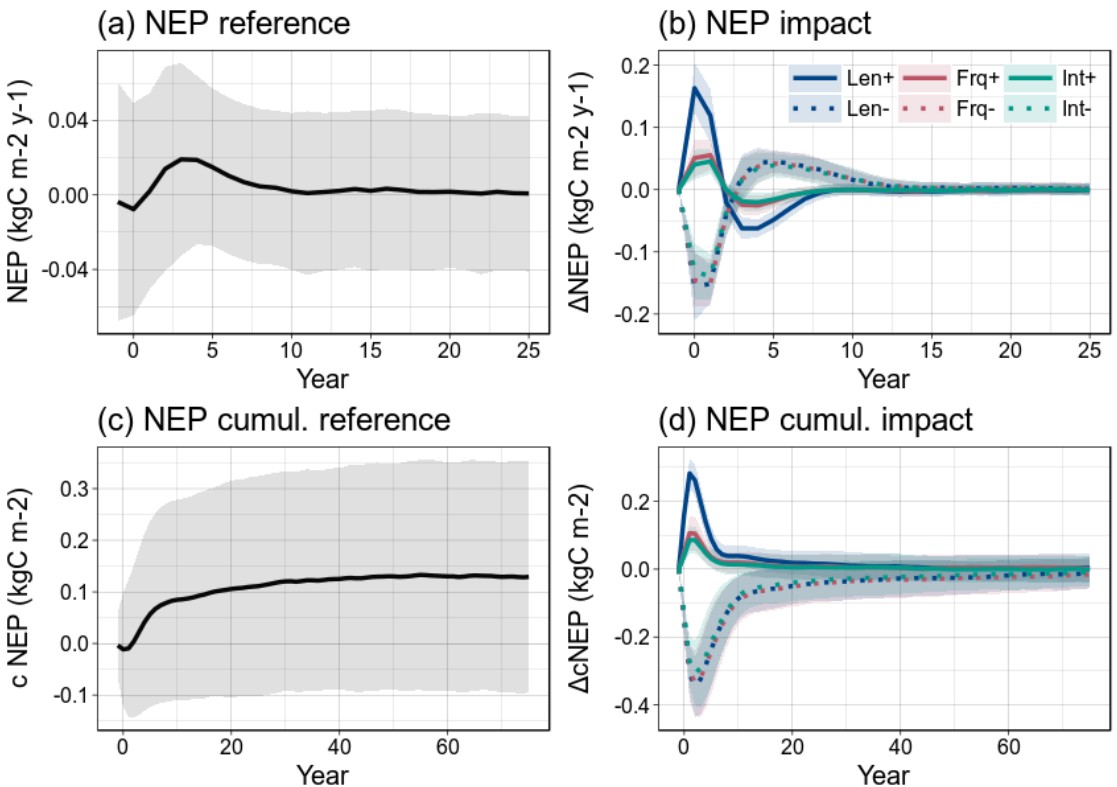

**Figure 9.** Impact of the different scenarios on the cumulative NEP at the Dahra site. (a) Average reference yearly NEP over a period of 25 years after the disturbance. (b) Impact of the disturbances on yearly NEP. (c) Average reference cumulative NEP on a longer time scale (70 years). (d) Impact of the disturbances on the cumulative NEP. The year prior to the

perturbations is used as a starting point for the cumulative sum. Shaded areas indicate variability of the model runs over all ensemble members ($\pm 1\sigma$).

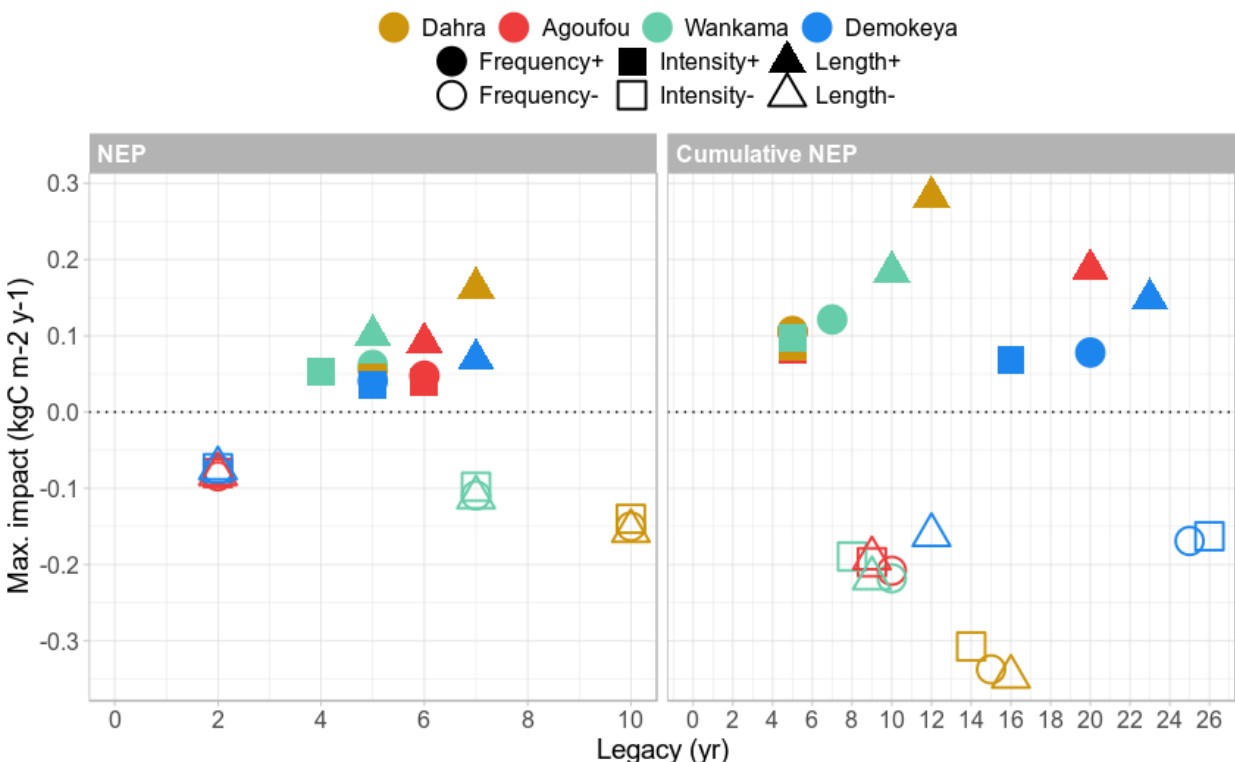

**Figure 10.** Summarized overview of the net ecosystem flux response to the different rainfall scenarios for all sites, showing the impact on the net ecosystem productivity (NEP) and the cumulative NEP. Different sites are represented by different colours, while the symbol shapes represent the applied scenarios.

The disturbance scenarios have a varying impact on the surface balance between water evaporation, runoff and infiltration into the soil (Fig. 11, 12). For disturbances that are based on a higher event frequency or a higher rainfall intensity, more than half of the added rainfall will be evaporated or lost to runoff for all sites, so that the resulting amount of infiltrated water will be reduced accordingly (Fig. 12). Increased rainfall intensity caused a ~10% higher loss through evaporation than the scenario of increased event frequency, while increased frequency caused a ~10% higher loss through runoff than rainfall intensity. For the scenario with an increased rainy season length, there is only a slight increase in runoff because the rainfall has more time to percolate through the soil before saturation occurs, while evaporation increased due to longer exposure. On the other hand, the scenarios with reduced precipitation all have similar impacts on surface water balance, again mainly varying between the different sites rather than the season characteristic (Fig. 12). Legacy was mostly site-dependent and timed around 3-4 years for positive impacts on evaporation and percolation, while the variation was higher for runoff (0-8 years). The legacy of reduced rainfall scenarios lasted again longer (3-10 years). During the years following the initial impact, the surface evaporation and runoff show a significant reversal in response at all sites. For the scenarios of reduced

rainfall, these overshoots are of the same order of magnitude as the initial impact (Fig. 11, S11-S13). For the Demokeya and
Wankama sites, the overshoots can be even higher than the initial impact, which is why the maximum impact value appears
in the (positive) upper half of Figure 12.

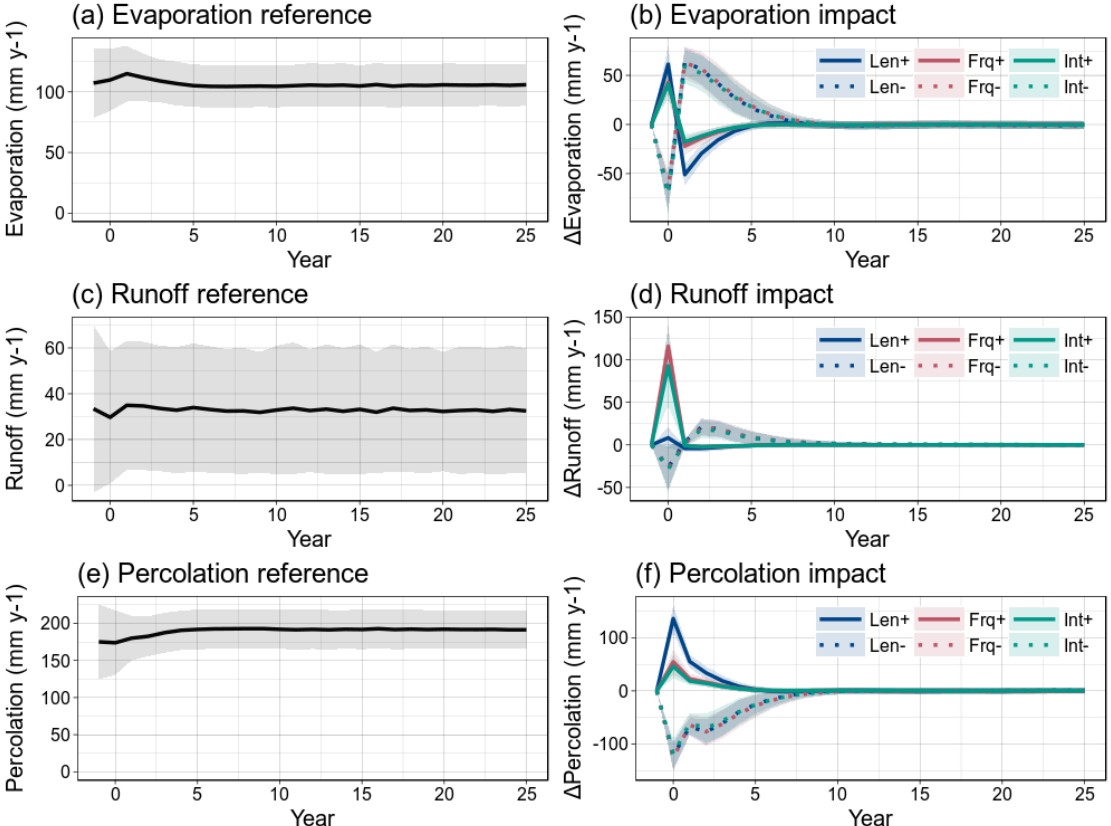

**Figure 11.** Impact of the different disturbance scenarios on surface water balance. Reference values and impact on (a,b)
surface evaporation, (c,d) surface runoff, and (e,f) infiltration of water into the soil. Shaded areas indicate variability of the
model runs over all ensemble members (±1σ). Results shown for the Dahra site simulations.

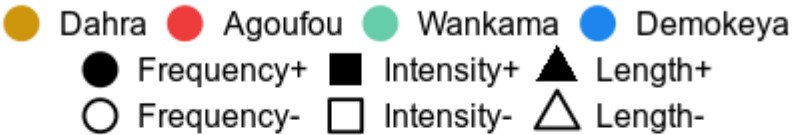

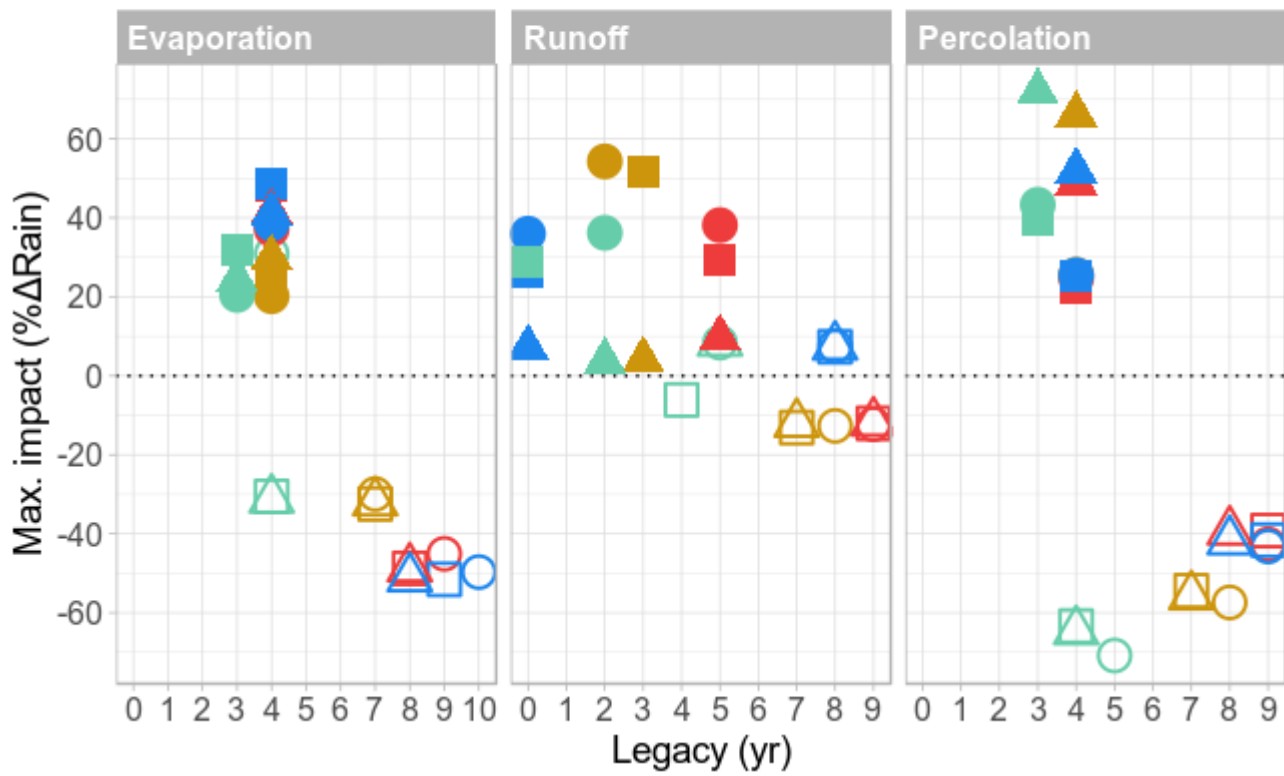

**Figure 12.** Impact of the different disturbance scenarios on surface water balance for all sites, showing the maximum impact on the surface evaporation, runoff and percolation as a percentage of added/reduced rainfall amount, along with its legacy (years).

Although fire can play a major role affecting the vegetation structure of African savannas, sites with a MAP below 350 mm are more rarely regulated by fire because of the low fuel availability (Sankaran et al., 2008). All sites considered in this study have a MAP of 339 mm or less (Table 1) and most fire events are anthropogenic. Nevertheless, as annual precipitation levels may increase under future climate scenarios, fires may play an increasing role at these sites in the future. The standard LPJ-GUESS model has a relatively simple fire module, where ignition is based on fuel load and litter moisture (Thonicke et al., 2001). In our study, fires only occurred at the wettest sites (Dahra and Wankama), when the fuel load was high and desiccated during a dry period following an occasional wet year. In these cases, fires mostly contributed for a small fraction (<3% of the GPP) to the total carbon emission from the ecosystem to the atmosphere, although there were a few exceptional

years where the contribution was higher (6-11%). This was also observed in the disturbance experiments: for the rainfall exclusion scenarios, fires increased during the disturbed year due to reduced moisture, adding on average 1.8 to 40 gC m$^{-2}$ year$^{-1}$ to the total carbon emissions, depending on the scenario. For rainfall addition scenarios there was a peak in fires in the year after the disturbance due to increased litter, adding up to 10 gC m$^{-2}$ year$^{-1}$ to the total carbon emissions. However, for all scenarios the model uncertainty of the carbon fluxes due to fires was high, as the standard deviation of the impacts over all ensemble members was similar to the mean impact.

## 4 Discussion

### 4.1 Summary of key results

A combination of the updated model parameter values and daily scale WFDEI-MSWEP meteorological drivers improved the agreement between model simulations and flux tower measurements of NEP for all Sahel sites. Our model disturbance experiments illustrated a strong contrast between scenarios of increased and decreased rainfall for all sites. For increased rainfall scenarios, the impact strongly depends on how rainfall is distributed over the season, while this was not the case for decreased rainfall scenarios. Out of the rainfall addition scenarios, increasing the length of the rainy season had the strongest impact at all sites, especially on $C_4$ grasses. Impact legacies are calculated conservatively, and generally last 1-5 years longer for negative scenarios compared to the positive scenarios, and are mostly site-dependent. The impact and legacy of scenarios of increased rainfall also varied among the sites. Grasses experienced their peak impact during the disturbed year and the legacy was limited to 3 years for positive scenarios, and 6 years for negative scenarios. For tree PFTs the peak impact was delayed by one year and it took longer to return to reference values. Impacts (positive or negative) on the net ecosystem carbon uptake (NEP) ranged from 10% to 47% of the reference yearly total photosynthesis (GPP), depending on the site and the applied disturbance. Due to overshoots following the initial NEP impact, the cumulative carbon uptake balances out again after a period which ranges from 6 to 25 years. The contrast between the scenarios is mainly explained by the different impacts on surface water balance, as increased season length allowed for a higher fraction of added rainfall to infiltrate into the soil than increased frequency or intensity, which led to a higher amount of runoff and surface evaporation respectively. The impact of rainfall reduction scenarios varied between the different sites, but not between the scenarios.

### 4.2 Model evaluation

The improved model performance was expected, as the published model PFTs (Smith et al., 2014) represent generic tropical species, while the new parameter values are specific for dryland ecosystems. A one-at-a-time sensitivity analysis revealed that updating other relevant parameters did not lead to a significant improvement in simulating the carbon fluxes and ET (not shown), but parameter covariance sensitivities are still to be tested. Moreover, the daily MSWEP data show a better match with precipitation measured at the flux tower (Fig. S3) and capture the onset and end of the rainy season better than the

interpolated CRU-NCEP monthly data. One factor that is not included in these simulations is livestock grazing, which was found to have a positive impact on both gross primary productivity (GPP) and ecosystem respiration (ER) (Tagesson et al., 2016b), although further studies are needed to fully understand this mechanism. At the Dahra site, cattle density was the highest during 2010, potentially explaining the higher discrepancy with the model that year (Fig. 5) (Tagesson et al., 2016b).

390 Simulated reference values of surface runoff are relatively high when compared against earlier published ranges for the Sahel (Fekete et al., 2002). This has been observed in earlier land surface model intercomparison studies as well, stressing the need for a good representation of soil hydrology in vegetation models (Grippa et al., 2017).

## 4.3 Response of woody versus herbaceous cover

At all sites, the model simulates co-existence of woody and herbaceous PFTs, which will compete for resources and 395 therefore generate complex vegetation dynamics. The herbaceous layer generally responded more strongly and swiftly to perturbations in precipitation than the woody vegetation for almost all sites and scenarios, especially to increases in rainy season length (Fig. 7-8). This contrasting behaviour reflects differences in plant representation in the model. Increased precipitation will lead to increased carbon uptake, which for grasses can be allocated to roots and leaves only. The tree PFTs will need to allocate a significant amount of carbon to woody components as well, which in turn will provide a safety net 400 during the scenarios of decreased rainfall. The difference in the timing of the impact between grasses and trees is also partly due to the differences in root distribution. Grasses will be mostly affected by the water content in the upper soil layer, as it contains 90% of their root biomass in the model. Therefore, they will directly respond to changes in precipitation (Brandt et al., 2018; Gherardi and Sala, 2015). In contrast, trees have 40% of their root biomass in the lower soil layer of the model, where the water content integrates changes in precipitation over a longer timeframe. Together with physiological differences 405 (e.g., allocation to woody parts), this explains the longer reaction time of the trees in the model. Further differences in response between evergreen and deciduous trees are due to their difference in SLA and phenology. Positive disturbances of increased rainfall initially benefit both woody PFTs, but the positive impacts last longer on evergreen trees, while the positive impacts on deciduous trees are followed by a negative overshoot, especially for the scenario of increased season length. Similarly, evergreen trees recover more slowly from negative disturbances than deciduous trees, which display a 410 positive overshoot following the initial negative impact. These results show that single-year disturbances can shift the weights in the competition for resources among the different PFTs for several years. Kulmatiski and Beard (2013) have shown experimentally that an increase in rainfall intensity (without changing the total rainfall) will increase aboveground woody plant growth and decrease grass growth. This behaviour is not observed in our model study. However, as Kulmatiski and Beard (2013) argued, this increase reflects the ability of woody plants to increase their rooting depth, a process that is 415 not included in our model, which only simulates two soil layers for which total root biomass can vary, but in which the PFT root distribution between the two layers remains fixed. Earlier research showed that the water use of *Acacia tortillis* trees in the Sahel is not much impacted by the dry season, as these trees have a deep taproot which reaches the water table (Do et al., 2008). Only after several dry years, when the water level in deeper soil layers plummets, this may have a major impact on

tree water stress. A new model version of LPJ-GUESS is being developed, which will contain 15 soil layers and therefore may address these issues more adequately in the future. The model does not contain complex dynamics such as hydraulic distribution, but Barron-Gafford et al. (2017) showed that the existence of taproots does not have a strong impact on tree-grass facilitation through hydraulic redistribution.

On the ecosystem scale there is an increase in carbon uptake in response to a year of increased precipitation, but most of this gain is quickly lost again during the following years (Fig. 9, S4). The largest part of the photosynthesised carbon will be allocated to leaf and root biomass of the C4 grasses, which will end up in the litter and soil carbon pools after the rainy season, where in turn most of it will respire again to the atmosphere during the following decades (Fig. S4). In response to a year of decreased precipitation, the ecosystem net productivity is reduced due to increased water stress on the plants, leading to a lower leaf production. During the years after this initial impact, the amount of leaf litter will therefore be lower than in the reference run. This causes a relative reduction in heterotrophic respiration and therefore a positive overshoot in the NEP impact. Nevertheless, it can take up to several decades before this increased NEP can balance out the initial carbon loss again (Fig. 9d, 10). For all scenarios and all sites, the ecosystem proved to be resilient to single-year disturbances in precipitation, as no permanent change in ecosystem state was induced. However, the high magnitude and extensive legacy of the impacts of a single-year disturbance on ecosystem carbon uptake may potentially drive the high contribution of drylands to inter-annual variability in the global land carbon sink (Ahlstrom et al., 2015; Poulter et al., 2014).

The contrast between the scenarios of increased rainfall was also simulated by Guan et al. (2018) for scenarios with long-term changes in precipitation, although only found for regions of higher mean annual precipitation (700-1600 mm year-1). Wu et al. (2018) reported a negative asymmetry in the response to exceptionally wet and dry years for temperate grasslands, where increases in aboveground net primary productivity (ANPP) were found to be smaller in magnitude during extremely wet years compared to decreases in ANPP during extreme dry years. The outcomes of our model experiments add a degree of nuance to these results, as we showed that the distribution of rainfall over the rainy season further modulates these asymmetrical responses: for the scenarios where rain event intensity or frequency were modified, the asymmetry was negative, while it was positive for scenarios in which season length was modified, i.e., increases in NPP due to a longer season had a larger magnitude than decreases due to a shorter season (Fig. S4). Wu et al. (2018) constructed their altered rainfall scenarios by increasing or decreasing the amount of rainfall in each event by a given factor, essentially producing scenarios of modified rainfall intensity (Supplementary Materials S1). Dannenberg et al. (2019) found that trees in drylands also displayed a negative asymmetry to precipitation variability, which was also further nuanced by our experiments. The contrast between the scenarios is explained by the amount of water that infiltrates the soil (Fig. 12, S11-S13), as the same asymmetry in the distribution of water between scenarios with increased and decreased rainfall was found in the vegetation response. These results agree with earlier research showing that soil texture and structure may play a mediating role in the vegetation response to rainfall variability, although in our study only the sandy soil type was used (Case and Staver, 2018). These results may also clarify why models better capture the response of vegetation to rainfall exclusion than addition in the study of Paschalis et al. (2020), as the impact of scenarios of decreased rainfall is much less dependent on the rainy season

characteristics than the impact of increased rainfall scenarios. Further vegetation model experiments could test the sensitivity of our results to different soil types, as we expect an even stronger contrast between the characteristics for finer structured soil. In general, local variations in hydraulic conductivity are known to have a major impact on the local-scale water balance in the Sahel, where soil surface crusting plays an important role (Leauthaud et al., 2017; Velluet et al., 2014).

It is expected that variations between the different sites are largely due to differences in historical meteorological conditions, as all other model parameters remained the same across sites. Variations between the sites were most clearly distinguished in the scenarios of reduced rainfall, where especially the impact legacy varied across the sites. No clear relationship was found between site conditions (Table 1) and ecosystem response to any of the disturbances. However, sites with a lower MAP, such as Demokeya and Agoufou, experienced a lower NEP impact and a shorter legacy from scenarios of reduced rainfall, compared to the wetter sites. The impact is likely lower because these drier sites will conceive a lower reference vegetation cover, leading to a lower impact on heterotrophic respiration after an exceptionally dry year. Similarly, wetter sites had a higher fraction of rainfall that percolated into the soil than drier sites, while surface evaporation increased more at drier sites than wetter sites. This may be due to shading, which is higher in wetter sites because of a higher vegetation cover. Nevertheless, in order to derive a clear relationship between site conditions (e.g. MAP) and disturbance impacts, a follow-up study could focus on sites along a stronger gradient in site conditions (e.g. a North-South precipitation gradient).

Finally, our results seem to contrast earlier research which has shown that phenology of cropland and grassland in Sub-Saharan Africa is mainly driven by photoperiodicity, while in our model a longer rainy season will cause a longer growing season (Adole et al., 2019). Photoperiodicity is only implemented for crop PFTs in LPJ-GUESS, while for the natural vegetation PFTs that were used in this study, simulated phenology is driven by water availability, and therefore follows the rainy season. However, at the local scale the importance of photoperiod is diminished in the Sahel. While individuals of several species are photoperiodic, phenological plasticity is strong and a longer rainy season does seem to bring a longer growing season due to cohort and species succession.

## 4.4 Strengths and limitations

The approach developed in this study presents a unique way to investigate the impact of different rainy season characteristics on the vegetation in the Sahel. Our algorithm allows us to create artificial rainfall scenarios which strongly resemble the original rainfall data, while also retaining the internal consistency with other meteorological variables. Some ensemble members in our scenarios may suffer from a loss of autocorrelation in temperature or incoming radiation, as the algorithm for constructing the artificial scenarios occasionally had to consult neighbouring pixels or longer time periods. However, as these dryland ecosystems are mostly sensitive to rainfall, and it was rainfall which was varied the strongest ($\pm 2\sigma$) in our simulations, we expect the influence of loss of autocorrelation to be limited. By introducing one single year of anomalous rainfall in the time series, the impact on the ecosystem can be quantified in both magnitude and timing. The use of large ensembles of such disturbance scenarios has enabled us to gain a detailed insight into the processes that drive this vegetation response in drylands. These techniques may be applicable in other regions and may be used to answer similar questions

related to climate sensitivity of ecosystems. While earlier research often focuses on the impact of intra-annual rainfall distribution and variability on tree cover, this work also takes into account the impact on grasses in drylands. Changes in ecosystem composition, such as woody encroachment, would manifest themselves over decadal timescales. Such changes are rather driven by changes in rainfall regimes than singular anomalous years, and therefore we do not expect such shifts to happen here. Applying extreme precipitation repeatedly over a variable number of consecutive years will potentially lead to a tipping point, after which the ecosystem does not recover to its original state. However, this is beyond the scope of this research and the question remains how realistic or useful such a threshold would be. Experimental verification of the results predicted by this model should be feasible, but the results of earlier in situ experimental studies (e.g., Kulmatiski and Beard, 2013) are difficult to link with this study. Finally, this study does not take into account possible asymmetries in the distribution of rainfall over the rainy season, as rainfall increases in the core wet season are found to impact herbaceous foliar mass, while increases in the early or late parts of the rainy season impact mainly woody foliage production (Brandt et al., 2019).

As this is a modelling study, its outcome is as reliable as the model assumptions and parameterization, in addition to the quality of the meteorological drivers. Allocation to carbon pools happens at the end of each simulated year, which may influence our results, although we mainly look at inter-annual impacts. A model version which includes daily carbon allocation for grasses has been developed for studying grass dynamics and grazing potential into more detail, which could be further developed to include daily allocation for tree PFTs as well (Boke-Olén et al., 2018). The representation of drought stress and hydraulic dynamics through the soil-plant-atmosphere continuum is expected to play an important role to determine the impact of drought response of ecosystems, especially in drylands (Medlyn et al., 2016). However, like many vegetation models, LPJ-GUESS has a relatively simple representation of these processes, based on empirical relationships (Gerten et al., 2004; Smith et al., 2014). Recently, efforts have been made to improve these processes in the Ecosystem Demography (ED) model, by including a representation of the hydraulic pathway through the plant, connecting phenology with hydraulic status, and by parameterizing the hydraulic model based on plant hydraulic traits (Xu et al., 2016). Adapting these or similar ideas for LPJ-GUESS will most likely improve both the validation and the reliability of the results presented in this study. The quality of soil hydrology representation in the model may have an influence on the results of this study as well, given the importance of runoff and percolation for the vegetation impact. Furthermore, implementing a photoperiodicity-driven phenology may be necessary to upscale this research to the regional level (Adole et al., 2019). Finally, although fires play a major role in regulating woody cover in African savannas, its impact was limited for the dryland sites in this study and the model uncertainty was high where fires occurred. Most likely fire will play a larger role at lower latitudes, where the MAP levels are sufficient to generate the necessary fuel load for fires to occur. Studying the impact of rainfall disturbances on fire occurrence in those regions will lead to a better understanding of the complex disturbance-driven dynamics of mesic savannas (Sankaran et al., 2008), especially when LPJ-GUESS is coupled with more sophisticated fire models such as SPITFIRE, which significantly improve the fire model performance in those regions (Thonicke et al., 2010).

*Author contribution.* W.V., G.S., S.H. and H.V. designed the research. W.V. performed model experiments and analysed the data. J.A., P.N.B, B.C., J.D., R.F., L.K. T.S. and T.T. collected the field data. W.V. drafted the paper and all authors contributed to writing the manuscript.

*Competing interests.* The authors declare that they have no conflict of interest.

*Acknowledgements.* The authors acknowledge the support for the U-TURN (Understanding Turning Points in Dryland Ecosystem Functioning, SR/00/339) project from the Belgian Federal Science Policy Office (BELSPO) in the framework of the STEREO III (Support to Exploitation and Research in Earth Observation) programme. Furthermore, the authors would like to sincerely thank Dr. M. Combe for the fruitful discussions, and M. Mbaye for the local support in Senegal.

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
