# Peer review of "Contrasting responses of woody and herbaceous vegetation to altered rainfall characteristics in the Sahel"

_Biogeosciences, 2020_

## Referee Comment (RC1) · Anonymous Referee #1 · 16 Jul 2020

This study used a model to investigate the impact of high and low precipitation years on ecosystem carbon and water cycling in the Sahel region, Africa. Annual precipitation anomalies were simulated by changing rainfall intensity, event duration, or rainy season length and holding the remaining two variables constant. The resultant fluxes were then analyzed with respect to contributions from the three major regional plant functional types. In this framework, grasses demonstrated a flashier response to all precipitation scenarios than trees, and changing the length of the rainy season had the greatest impact on vegetation due to changes in runoff and surface evaporation associated with precipitation intensity and duration. No single-year precipitation treatments had a significant effect on the long-term (decadal) simulated carbon or water fluxes.

[Figure]

General comments: I thought this research was creative and well designed, and work from understudied regions like Africa is especially valuable. However, the stated research questions (L91-93) are left mostly unanswered by the study that presents data from only one site in its current form.

I don't understand or agree with the decision to exclude three of the four sites from the main manuscript text. I realize that these results are available in the supplement (without interpretation or discussion), but they're needed in the main text insofar as they're key to meaningful interpretation of your work and its scope. To me, the impact of this manuscript, and thus the suitability for the journal, depends on its applicability to other dryland systems. To establish this, (1) model performance must be evaluated at all sites and (2) systematic biases (or lack thereof) subsequently analyzed with respect to relevant climate/soil/vegetation, etc. characteristics, in order to generate transferable biogeoscientific insight.

Specific comments: L16-19: Are extremely high precipitation years also forecast for the Sahel? Only drought extremes are invoked on L17.

L20: "The rainy season" is confusing as you have yet to establish precipitation seasonality. Also "signature" could be more specific e.g., "the impact of the rainy season on the ecosystem carbon balance" or similar.

L24: Please clarify what's meant by "meteorological consistency" in this context. Reading ahead, I see it now, but if you wish to invoke this term/concept in the abstract, it should stand alone.

L29: Is it your intention to use "semi-arid" and "dryland" interchangeably? Maybe better to just choose one term and stick with it, at least in the abstract.

L30: After reading the entire manuscript, I disagree with the use of "long-term" here.

L54: And presumably in other dryland areas right? It's to your advantage to keep the results as broadly applicable as possible.

L71-72: Shouldn't mean annual precipitation (MAP) be less in arid areas than in semi-arid areas? Also 650 mm described as the cutoff for mesic (L45) but then 700 mm cited as a cutoff for "less arid" (L73). Please establish a clear precipitation classification scheme.

Table 1: Does "fallow bush" imply 0% tree cover at the Niger site?

L103: Most readers will be unfamiliar with this area and would benefit from a site map.

L108: I think including the measurement years for each of the flux tower sites would be warranted here or in Table 1. They're used to validate the model, so the length of the instrumental record is particularly relevant. Also it bothers me when eddy covariance measurements are simply presented as "the truth"; please indicate at minimum the mean uncertainty associated with the flux observations.

L140-141: This is unexpected given "soil control on surface water balance" (L28) and the different soil classifications in Table 1.

L158: How was the length (1 to 37 years) of any particular meteorological cycle determined?

L160: Just the rainy season or the entire year (as implied by the Figure 1 y-axis)? This may seem common knowledge to you, but most readers will be unfamiliar with the climatology of the Sahel. Maybe a "percent of MAP during the rainy season" metric could be added to Table 1?

Figure 1: I'm unsure how to interpret the frequency variable; would a value of 0.5 day-1 correspond to one rain event every two days? Perhaps an explanatory sentence could be added to the caption (or just a pointer to Table 3)? How was the length of the rainy season determined quantitatively? I'm more familiar with growing season length calculations that can greatly affect interpretation of results. Reading ahead I see a definition in terms of "climatological anomalous accumulation" in Table 3, does this imply a typical lack of any precipitation outside of the rainy season? If so, please

state clearly in the site description.

Table 4: It's taken me a while to interpret the "rainfall disturbance" column. I think it means the net change in rainy season precipitation due to perturbation of one of the three precipitation metrics. Perhaps change to "change in rainy season precipitation" or similar to clarify this? And/or list the relevant perturbation first and then the resulting effect second?

Figure 3: Can you speculate as to why there is a much larger difference between modeled and observed data at Dahra in 2010 compared to any other year? This could inform your other results (see general comments).

Figure 4: Many people (including myself) find Taylor diagrams difficult to interpret. Why not just show an analog to Figure 3 for all three sites? This would provide ecohydrological information about each site, as well as highlight periods of relative agreement and disagreement between the modeled and observed data i.e., my previous comment, which could yield additional insights. Why RMSE on the previous figure and RMSD here?

L231: I'm unclear as to how exactly +122% and +54% are meant to be interpreted. For example, is it that increased rainy season length stimulated the C4 grass LAI 122% more than both similarly increased frequency and intensity? Perhaps including some of the actual values would clarify this.

Figure 5: How to interpret (a-c)? Is it simply the aggregate of (d-f)? If so, what information is meant to be taken away from (a-c)? What about the other three sites? Given that this is a recurring framework, please treat this as a major comment.

L253-254: I see no mention of autotrophic respiration, do you mean GPP? How was it calculated? See Chapin et al. 2006 Reconciling Carbon-Cycle Concepts, Terminology, and Methods for relevant definitions.

L255: Excluding changes in respiration, how can it be less than 1x the reference NPP

if there's additional photosynthesis?

Figure 6: Same comment as Figure 5.

L272-276: Suggest rearranging the text (or figure) to introduce figure panels in order.

L287: Four sites were introduced in the methods, but so far, we've only seen results from Dahra. This is confusing and requires explanation and justification for your rationale.

L290-292: This harkens back to my comment on L140-141, perhaps the effect of soil type/texture on the results is discussed in the next section. . .

L298-302: Right on cue, here's a partial response to my comment on L287. In short, I'm not satisfied with your treatment of the other sites. The vast majority of readers will not reference the SI, and even those that do must have model interpretation expertise to gain useful information. This functionally excludes three of the four sites from your analysis. Without these other sites, you're left with a site-specific tuned model that will be of little interest to other dryland or even Sahel researchers. To say it a different way, in my eyes, the strength of your work is the development of a transferrable model that could be used by dryland savanna researchers globally.

This doesn't have to be a fatal flaw as I think you have the data to support a broadly applicable presentation/contribution, they just have to be worked into the manuscript. Especially in light of my comments about the "reference" panels in Figures 5-7, I think you have the space for this. And it's okay if the model performance wasn't as good at the other sites (not saying this is the case), simply presenting the data and discussing potential reasons for periods of better and worse model/data agreement would speak to process-based information and benefit the research community.

L317: You've discussed both NPP and NEP, so the "total carbon flux" is unclear. Also fires emit carbon dioxide, so do you mean to convey that they reduce (not add to) the carbon flux by this amount?

L326-328: Suggest adding an initial discussion paragraph before section 4.1 that summarizes the principal results and why they matter. This is particularly important given the lack of a conclusion section.

355-356: Is there evidence for hydraulic redistribution in these Acacias? If so, the potential for impacts on grass-tree facilitation vs competition must be addressed.

367-368: This is an important result that could elevated to the abstract and/or first discussion paragraph.

L372: Not currently corroborated at all four sites.

L374-376: Recent work by Dannenberg et al. also showed negative asymmetry (with respect to precipitation variability) for trees in semiarid areas:

MP Dannenberg et al. 2019. Reduced tree growth in the semiarid United States due to asymmetric responses to intensifying precipitation extremes. Science Advances.

L389-392: Some kind of sensitivity analysis would go a long way toward reducing the uncertainty associated with this caveat e.g., a comparison of results from model runs where the texture was varied by increments of ∼5% within some permissible range.

L393: "Variations between the different sites" have not been shown.

L402-403: A citation would strengthen this claim.

---

## Referee Comment (RC2) · Anonymous Referee #2 · 28 Jul 2020

The manuscript presents a modelling study quantifying the impact of rainfall amounts, as well as its intra-annual variability on semi-arid ecosystem responses, including gross and net ecosystem carbon fluxes, as well as water fluxes (evapotranspiration and runoff). Overall, the study is well put together, but some key clarifications are needed:

a) Model selection a1 - Model selection is of paramount importance in a modelling study as such. The authors should motivate why they opted for LPJ-GUESS. Semiarid ecosystems as the ones analysed here are very sensitive to plant water availability, and the vertical distribution of soil moisture within the root zone will play a key role. Why did the authors choose a model that does not explicitly resolve the Richards equations? A

large fraction of terrestrial ecosystem models does that. a2 – A better representation of the semi-arid vegetation was implemented by altering two traits, the SLA and wood density in pre-existing PFTs. However, a crucial "trait" in such ecosystems in drought deciduousness. Can the authors provide more information on how this was implemented? a3 – L116: The model resolves carbon dynamics, and allocation to carbon pools at the end of the year. As the authors look at intra-annual variability of vegetation responses, that might be problematic, as same year carbon dynamics, might be mis-represented. Also resolving carbon dynamics at the end of the year, might impact the ecosystem legacies. Can the authors further elaborate on this potential limitation?

b) Data b1 – Why did the authors opted for the ERA-Interim reanalysis data, and not for the more recent and better ERA5, which is also available at a much finer spatial resolution. b2- To my understanding, when pooling data from observed days for the synthetic time series, temporal autocorrelation is not conserved. Is that true? If yes what is the potential impact on the results (i.e. lack of strong correlation of temperature – i.e. long-lasting heatwaves etc).

c) Results c1 – To my understanding the model predicts species coexistence in all sites. Such co-existence, will most likely affect the decadal long legacies presented here. Is that in agreements with what is observed at each of the sites? Does s[ecies co-existence occur in reality? c2 – While legacies in drought responses have been widely observed, previous studies (e.g. Kolus et al., 2019, Scientific reports, doi:s41598-019-39373-1 ) have found that terrestrial ecosystem models underestimate them. Can the authors explain why in their results decade long-lasting legacies (typically longer than observed) occur? Is that primarily due to disturbances? c3 – I agree with reviewer 1 regarding the use of the Taylor diagrams. My main disagreement on their interpretation is that due to the high seasonality of the climate, most of the correlation comes from being able to reproduce the annual cycle, and not reflecting the performance of the model regarding rainfall structure. Possibly a Taylor diagram performed at e.g. seasonal anomalies would be more informative.

d) Presentation I fully agree with reviewer 1, regarding the choice of the authors to present one site and append in the supplementary the analysis of the remaining three. In fact, a detail comparison of the four sites would significantly strengthen the results and provide further mechanistic insights regarding ecosystem functioning.
* * *

---

## Author Comment (AC1) · 4 Sep 2020

**Response to Anonymous Referee #1**

We would like to thank both Anonymous Referees for their review and suggestions for improvements.

This study used a model to investigate the impact of high and low precipitation years on ecosystem carbon and water cycling in the Sahel region, Africa. Annual precipitation anomalies were simulated by changing rainfall intensity, event duration, or rainy season length and holding the remaining two variables constant. The resultant fluxes were then analyzed with respect to contributions from the three major regional plant functional types. In this framework, grasses demonstrated a flashier response to all precipitation scenarios than trees, and changing the length of the rainy season had the greatest impact on vegetation due to changes in runoff and surface evaporation associated with precipitation intensity and duration. No single-year precipitation treatments had asignificant effect on the long-term (decadal) simulated carbon or water fluxes.

Response: We would like to clarify that we did not study the anomalies caused by changing event duration, i.e. the average number of consecutive rainy days, but rather the event frequency, i.e. the average number of consecutive dry days between rain events. Of course, these are related when the length of the rainy season is kept constant.

**General comments**

I thought this research was creative and well designed, and work from understudied regions like Africa is especially valuable. However, the stated research questions (L91-93) are left mostly unanswered by the study that presents data from only one site in its current form. I don't understand or agree with the decision to exclude three of the four sites from the main manuscript text. I realize that these results are available in the supplement (without interpretation or discussion), but they're needed in the main text insofar as they're key to meaningful interpretation of your work and its scope. To me, the impact of this manuscript, and thus the suitability for the journal, depends on its applicability to other dryland systems. To establish this, (1) model performance must be evaluated at all sites, and (2) systematic biases (or lack thereof) subsequently analyzed with respect to relevant climate/soil/vegetation, etc. characteristics, in order to generate transferable biogeoscientific insight.

Response: We thank the reviewer for her/his appreciation for our study design and focus on the Sahel region. We agree that it would be better to analyse the results of all four study sites in the main text, as this can indeed further contribute to the knowledge on how global drylands respond to extremes in seasonal characteristics. We initially chose to focus on one site in the main text, mainly to make the manuscript not too lengthy.

We hope that it would be a good compromise to include a few figures which summarize the results from all sites, only showing the maximum amplitude and the legacy for each scenario. Such a figure is given below, where we summarize the scenario impacts on leaf area index (LAI) for each PFT (Figure R1). As it can be seen, there are some contrasts in both amplitude and legacy between the different sites and we will elaborate further on these differences in the Results and Discussion sections. Of course, such a summarised presentation has its limitations, as for example overshoots are neglected, but the full time-series can still be found in the supplementary materials.

**Figure R1.** Absolute amplitude and legacy of the vegetation response to (a) positive and (b) negative rainfall perturbations due to (a) increased and (b) decreased event frequency, intensity or season length. The legacy was defined as the last year for which the ensemble average (absolute) response is larger than the ensemble standard deviation, while the amplitude is the maximum response, relative to the reference run values. Results shown for all Sahel fluxtower sites.

**Specific comments**

L16-19: Are extremely high precipitation years also forecast for the Sahel? Only drought extremes are invoked on L17.

Response: We will look into this again and add relevant references to the Introduction.

L20: "The rainy season" is confusing as you have yet to establish precipitation seasonality. Also "signature" could be more specific e.g., "the impact of the rainy season on the ecosystem carbon balance" or similar.

Response: We will rephrase this line so that it will be more clear.

L24: Please clarify what's meant by "meteorological consistency" in this context. Reading ahead, I see it now, but if you wish to invoke this term/concept in the abstract, it should stand alone.

Response: We will remove this statement, as it is not of central importance to mention this in the abstract.

L29: Is it your intention to use "semi-arid" and "dryland" interchangeably? Maybe better to just choose one term and stick with it, at least in the abstract.

Response: Agreed and the manuscript will be modified accordingly.

L30: After reading the entire manuscript, I disagree with the use of "long-term" here.

Response: We agree that this term can be interpreted in many ways, depending on the background of the reader. Our (unwritten) definition of long-term was "more than an order of magnitude longer than the duration of the disturbance", in which case its use is justified. Nevertheless, we will clarify this sentence so that it is less open for interpretation.

L54: And presumably in other dryland areas right? It's to your advantage to keep the results as broadly applicable as possible.

Response: Agreed.

L71-72: Shouldn't mean annual precipitation (MAP) be less in arid areas than in semi-arid areas? Also 650 mm described as the cutoff for mesic (L45) but then 700 mmcited as a cutoff for "less arid" (L73). Please establish a clear precipitation classification scheme.

Response: We used the definitions from the respective papers here, but we agree that it would be good to translate them into a consistent precipitation classification scheme for our manuscript, and we will do so accordingly.

Table 1: Does "fallow bush" imply 0% tree cover at the Niger site?

Response: No, this fallow site does contain a few trees, but the dominant vegetation is a mixture of shrubs (mainly Guiera senegalensis) and annual grasses (Boulain et al., 2009). We will clarify this in the table.

L103: Most readers will be unfamiliar with this area and would benefit from a site map.

Response: Agreed and will be added.

L108: I think including the measurement years for each of the flux tower sites would bewarranted here or in Table 1. They're used to validate the model, so the length of the instrumental record is particularly relevant. Also it bothers me when eddy covariance measurements are simply presented as "the truth"; please indicate at minimum the mean uncertainty associated with the flux observations.

Response: Agreed. The measurement years will be added to the table, and the uncertainty on the eddy covariance measurements will be quantified and discussed.

L140-141: This is unexpected given "soil control on surface water balance" (L28) and the different soil classifications in Table 1.

Response: All of the sites consist of Arenosol, which are all sandy soil types. The model does not take into account differences in lower-level soil classification (i.e. Luvic, Ferralic or Cambic). We will add this information to the discussion.

L158: How was the length (1 to 37 years) of any particular meteorological cycle determined?

Response: The input meteorological dataset consists of 37 years (1979-2016). We marked all extreme (dry or wet) years and perturbed the remaining years one by one, represented by the different branches marked as "experiments" in Figure 2. Each branch is simulated until the year that is perturbed, and is then followed by a spin-down run.

L160: Just the rainy season or the entire year (as implied by the Figure 1 y-axis)? This may seem common knowledge to you, but most readers will be unfamiliar with the climatology of the Sahel. Maybe a "percent of MAP during the rainy season" metric could be added to Table 1?

**Response: Indeed, we have not mentioned that the climate in the Sahel has a high degree of seasonality. We will elaborate on this and add a seasonality metric in the table or in the main text.**

Figure 1: I'm unsure how to interpret the frequency variable; would a value of 0.5 day-1 correspond to one rain event every two days? Perhaps an explanatory sentence could be added to the caption (or just a pointer to Table 3)? How was the length of the rainy season determined quantitatively? I'm more familiar with growing season length calculations that can greatly affect interpretation of results. Reading ahead I see a definition in terms of "climatological anomalous accumulation" in Table 3, does this imply a typical lack of any precipitation outside of the rainy season? If so, please state clearly in the site description.

Response: The frequency characteristic is the inverse of the number of days between two rain events, regardless of how long these events last. There might be a small amount of rainfall outside the season, but the climate has a high degree of seasonality.

Table 4: It's taken me a while to interpret the "rainfall disturbance" column. I think it means the net change in rainy season precipitation due to perturbation of one of the three precipitation metrics. Perhaps change to "change in rainy season precipitation" or similar to clarify this? And/or list the relevant perturbation first and then the resulting effect second?

**Response: We will rearrange the table columns and rename "Rainfall disturbance".**

Figure 3: Can you speculate as to why there is a much larger difference between modeled and observed data at Dahra in 2010 compared to any other year? This could inform your other results (see general comments).

Response: One factor that is not included in our model simulations is livestock grazing, which was found to have a positive impact on both gross primary productivity (GPP) and ecosystem respiration (ER), although further studies are needed to fully understand this mechanism (Tagesson et al., 2016). Cattle density was the highest during 2010, potentially explaining the higher discrepancy with the model that year (Tagesson et al., 2016). We will elaborate further on the differences between modelled and measured fluxes, and their potential causes, in the revised manuscript.

Figure 4: Many people (including myself) find Taylor diagrams difficult to interpret. Why not just show an analog to Figure 3 for all three sites? This would provide ecohydrological information about each site, as well as highlight periods of relative agreement and disagreement between the modeled and observed data i.e., my previous comment, which could yield additional insights. Why RMSE on the previous figure and RMSD here?

Response: Taylor diagrams provide a compact way to directly visualize an evaluation of model performance, based on different metrics. In this diagram we evaluate the results for four sites, each based on three different meteorological drivers, which would lead to at least eight time-series plots if we include evapotranspiration as well. We refrained from showing all these in the main text due to space limitation. However, as noted by Anonymous Referee #2, the current diagram mainly evaluates how well the model captures the seasonality of the growing season, rather than the fluxes during the season. Therefore we will add a second Taylor diagram based on the growing season separately, and we will make sure to use consistent terminology throughout the manuscript (ie. RMSE).

L231: I'm unclear as to how exactly +122% and +54% are meant to be interpreted. For example, is it that increased rainy season length stimulated the C4 grass LAI 122% more than both similarly increased frequency and intensity? Perhaps including someof the actual values would clarify this.

Response: All relative impacts are given with respect to the reference runs, see next point.

Figure 5: How to interpret (a-c)? Is it simply the aggregate of (d-f)? If so, what information is meant to be taken away from (a-c)? What about the other three sites? Given that this is a recurring framework, please treat this as a major comment.

Response: Plots (a-c) represent the reference runs, or "control runs" if this term would be more clear. Each of these reference runs, i.e. each ensemble member, is driven by exactly the same meteorological sequence as its partner experiment run, but without the perturbation applied. In other words, these plots show what the LAI of the PFTs would be if the anomalous year did not take place. Plots (d-f) then show the relative difference with these runs, i.e. the absolute difference divided by the reference value. Therefore, showing these reference values is not only justified, but necessary in order to correctly interpret the impacts shown in (d-f). This answers the previous question on L231 as well.

L253-254: I see no mention of autotrophic respiration, do you mean GPP? How was it calculated? See Chapin et al. 2006 Reconciling Carbon-Cycle Concepts, Terminology, and Methods for relevant definitions.

*Response: We did mean the NPP, although we indeed didn't mention autotrophic respiration in the text. We will clarify this.*

L255: Excluding changes in respiration, how can it be less than 1x the reference NPP if there's additional photosynthesis?

Response: All results given here are relative to the reference value. The cumulative impact (d-f) is obtained as follows. First the impact on the yearly NPP is calculated by subtracting the reference yearly NPP from the total NPP. Then the cumulative sum is taken over the resulting values, to incorporate respiration changes as well. We agree that the manuscript may benefit from further clarifications here, and we will add them accordingly.

Figure 6: Same comment as Figure 5.

Response: See previous answers.

L272-276: Suggest rearranging the text (or figure) to introduce figure panels in order.

Response: Agreed, we will rearrange the text.

L287: Four sites were introduced in the methods, but so far, we've only seen resultsfrom Dahra. This is confusing and requires explanation and justification for your rationale.

**Response: See response to the general comments above.**

L290-292: This harkens back to my comment on L140-141, perhaps the effect of soiltype/texture on the results is discussed in the next section...

Response: This falls outside the scope of this study, as the different sites used the same soil type in the model. In the manuscript we already discuss that a sensitivity test on different soil types may be an interesting follow-up research paper.

L298-302: Right on cue, here's a partial response to my comment on L287. In short, I'm not satisfied with your treatment of the other sites. The vast majority of readers will not reference the SI, and even those that do must have model interpretation expertise to gain useful information. This functionally excludes three of the four sites from your analysis. Without these other sites, you're left with a site-specific tuned model that will be of little interest to other dryland or even Sahel researchers. To say it a different way,in my eyes, the strength of your work is the development of a transferrable model that could be used by dryland savanna researchers globally. This doesn't have to be a fatal flaw as I think you have the data to support a broadly applicable presentation/contribution, they just have to be worked into the manuscript. Especially in light of my comments about the "reference" panels in Figures 5-7, I think you have the space for this. And it's okay if the model performance wasn't as good at the other sites (not saying this is the case), simply presenting the data and discussing potential reasons for periods of better and worse model/data agreement would speak to process-based information and benefit the research community.

*Response: The results from the other sites will be brought into the main manuscript, see response to the general comments above.*

L317: You've discussed both NPP and NEP, so the "total carbon flux" is unclear. Also, fires emit carbon dioxide, so do you mean to convey that they reduce (not add to) the carbon flux by this amount?

Response: We agree that the terminology is a bit mixed up and leaves too much room for interpretation, for example regarding which convention of positive flux is used. Indeed, fires of course reduce the net productivity by adding to the emission of carbon. We will clarify this accordingly in the text.

L326-328: Suggest adding an initial discussion paragraph before section 4.1 that summarizes the principal results and why they matter. This is particularly important given the lack of a conclusion section.

**Response: Agreed, we will add this.**

L355-356: Is there evidence for hydraulic redistribution in these Acacias? If so, the potential for impacts on grass-tree facilitation vs competition must be addressed.

Response: We are not aware of hydraulic redistribution in these trees, but we can shortly discuss this phenomenon and its potential impacts. In either case, our vegetation model is not able to simulate such detailed plant hydraulics.

367-368: This is an important result that could elevated to the abstract and/or first discussion paragraph.

Response: Agreed, we will add it.

L372: Not currently corroborated at all four sites.

Response: We will discuss the results of the other sites into more detail.

L374-376: Recent work by Dannenberg et al. also showed negative asymmetry (withrespect to precipitation variability) for trees in semiarid areas:MP Dannenberg et al. 2019. Reduced tree growth in the semiarid United States due asymmetric responses to intensifying precipitation extremes. Science Advances.

Response: Thank you, we will look into this paper and add it to the discussion.

L389-392: Some kind of sensitivity analysis would go a long way toward reducing the uncertainty associated with this caveat e.g., a comparison of results from model runs where the texture was varied by increments of  $\sim$  5% within some permissible range.

Response: Indeed, this would be a highly interesting topic for a follow-up study. In the current manuscript we already discuss several dimensions of variation, so including changes in soil texture would add even more results that need to be presented.

L393: "Variations between the different sites" have not been shown.

Response: We will discuss the results of the other sites, and the variations between them, into more detail (see previous answers).

L402-403: A citation would strengthen this claim

Response: This claim follows from a discussion with field-experts. As they are included as coauthors, this claim is for now given as such, but we will try to find relevant papers and add them.

**References**

Boulain, N., Cappelaere, B., Ramier, D., Issoufou, H. B. A., Halilou, O., Seghieri, J., Guillemin, F., Oï, M., Gignoux, J. and Timouk, F.: Towards an understanding of coupled physical and biological processes in the cultivated Sahel - 2. Vegetation and carbon dynamics, J. Hydrol., 375(1–2), 190–203, doi:10.1016/j.jhydrol.2008.11.045, 2009.

Tagesson, T., Ardö, J., Guiro, I., Cropley, F., Mbow, C., Horion, S., Ehammer, A., Mougin, E., Delon, C., Galy-Lacaux, C. and Fensholt, R.: Very high CO2 exchange fluxes at the peak of the rainy season in a West African grazed semi-arid savanna ecosystem, Geogr. Tidsskr. - Danish J. Geogr., 116(2), 93–109, doi:10.1080/00167223.2016.1178072, 2016.

---

## Author Comment (AC2) · 4 Sep 2020

**Response to Anonymous Referee #2**

*We would like to thank both Anonymous Referees for their review and suggestions for improvements.*

The manuscript presents a modelling study quantifying the impact of rainfall amounts, as well as its intra-annual variability on semi-arid ecosystem responses, including gross and net ecosystem carbon fluxes, as well as water fluxes (evapotranspiration and runoff). Overall, the study is well put together, but some key clarifications are needed:

*Response: We thank the reviewer for her/his positive overall appreciation.*

**a) Model selection**

Model selection is of paramount importance in a modelling study as such. The authors should motivate why they opted for LPJ-GUESS. Semiarid ecosystems as the ones analysed here are very sensitive to plant water availability, and the vertical distribution of soil moisture within the root zone will play a key role. Why did the authors choose a model that does not explicitly resolve the Richards equations? A large fraction of terrestrial ecosystem models does that.

*Response: The use of LPJ-GUESS for this study mainly follows from practical reasons, as there was already a great expertise of running and developing the model in our group. In addition, the model is known to give a reasonable representation of large-scale sensitivities to drought in drylands at the global scale, and for Africa specifically (Ahlstrom et al., 2015; Brandt et al., 2017, 2018). It is our ambition to use this model at a larger scale for upcoming studies. We will add this motivation more clearly in the introduction. As already written in the Discussion section, we agree that dryland model performance may benefit greatly from using a model which resolves soil hydrology and plant hydraulics with a higher degree of detail. Therefore we are currently parameterizing and running the latest version of the ED2 dynamic vegetation model (Longo et al., 2019; Xu et al., 2016) for these sites.*

A better representation of the semi-arid vegetation was implemented by altering two traits, the SLA and wood density in pre-existing PFTs. However, a crucial "trait" in such ecosystems is drought deciduousness. Can the authors provide more information on how this was implemented?

*Response: Phenology of the deciduous PFTs is based on a water stress scalar ($\omega$) in the model. Low values of this scalar represent stress due to reduced soil water content, leading to a reduction of photosynthesis through stomatal closure. When this variable drops below a given threshold ($\omega_{min}$), the dry season starts and deciduous trees will shed their leaves. Likewise, when this scalar rises above $\omega_{min}$ new leaves will be produced, taking into account a prescribed minimum dormancy period (Smith et al., 2014). This further elaboration on drought deciduousness will be added to the Methods section.*

L116: The model resolves carbon dynamics, and allocation to carbon pools at the end of the year. As the authors look at intra-annual variability of vegetation responses, that might be problematic, as same year carbon dynamics, might be misrepresented. Also resolving carbon dynamics at the end of the year, might impact the ecosystem legacies. Can the authors further elaborate on this potential limitation?

*Response: Yes, we will elaborate on this in the discussion section. A model version which includes daily carbon allocation for grasses has been developed for studying grass dynamics and grazing impacts within a year (Boke-Olén, 2017). Comparing the outcomes of both model versions could be an interesting topic for a follow-up paper.*

**b) Data**

Why did the authors opted for the ERA-Interim reanalysis data, and not for the more recent and better ERA5, which is also available at a much finer spatial resolution.

*Response: We used reanalysis data based on ERA-Interim for the temperature and incoming shortwave radiation data only (Weedon et al., 2014). However, the most important driver in drylands is precipitation, for which we used Multi-Source Weighted-Ensemble Precipitation (MSWEP) data, which performed best in two large-scale evaluations that included ERA5 data as well (Beck et al., 2017, 2019).*

To my understanding, when pooling data from observed days for the synthetic time series, temporal autocorrelation is not conserved. Is that true? If yes what is the potential impact on the results (i.e. lack of strong correlation of temperature– i.e. long-lasting heatwaves etc).

*Response: As discussed in section 2.5, loss of autocorrelation was avoided by restricting the data resampling to stay as close in time as possible to the original DOY, thereby retaining synoptic patterns as much as possible. As it was not always possible to find a day with an amount of rainfall that matched the goal, neighbouring pixels or years may be consulted. In the latter case, temporal autocorrelation will likely be lost. We will elaborate on this potential limitation in the discussion.*

**c) Results**

To my understanding the model predicts species coexistence in all sites. Such co-existence, will most likely affect the decadal long legacies presented here. Is that in agreements with what is observed at each of the sites? Does species co-existence occur in reality?

*Response: Yes, our sites consist of a sparse woody cover in a herbaceous matrix (Table 1). The model indeed predicts coexistence of plant functional types at all sites as well. This co-existence will indeed have an influence on our results through the competition for resources. We will elaborate further on this in the discussion.*

While legacies in drought responses have been widely observed, previous studies (e.g. Kolus et al., 2019, Scientific reports, doi:s41598-019-39373-1) have found that terrestrial ecosystem models underestimate them. Can the authors explain why in their results decade long-lasting legacies (typically longer than observed) occur? Is that primarily due to disturbances?

*Response: The long legacies in our results originate mainly from the water content of the lower soil layer and the allocation of woody biomass. These pools respond relatively slow, compared to the timeframe of the simulated disturbances. Empirical observations of the disturbance impact legacies are lacking for drylands, so it is hard to evaluate our results against current data. Differences in model parameterization (i.e. dryland specific or not) and experimental setup may explain contrasts with earlier studies such as Kolus et al. (2019).*

I agree with reviewer 1 regarding the use of the Taylor diagrams. My main disagreement on their interpretation is that due to the high seasonality of the climate, most of the correlation comes from

being able to reproduce the annual cycle, and not reflecting the performance of the model regarding rainfall structure. Possibly a Taylor diagram performed at e.g. seasonal anomalies would be more informative.

*Response: Agreed. We will add a second Taylor diagram which only evaluates the model during the growing season.*

**d) Presentation**

I fully agree with reviewer 1, regarding the choice of the authors to present one site and append in the supplementary the analysis of the remaining three. In fact, a detail comparison of the four sites would significantly strengthen the results and provide further mechanistic insights regarding ecosystem functioning.

*Response: See the response to Anonymous Referee #1 on this issue. In short, we will add summarized results for the other sites to the main text, together with a more detailed comparison in the discussion.*

**References**

*Ahlstrom, A., Raupach, M. R., Schurgers, G., Smith, B., Arneth, A., Jung, M., Reichstein, M., Canadell, J. G., Friedlingstein, P., Jain, A. K., Kato, E., Poulter, B., Sitch, S., Stocker, B. D., Viovy, N., Wang, Y. P., Wiltshire, A., Zaehle, S. and Zeng, N.: The dominant role of semi-arid ecosystems in the trend and variability of the land CO2 sink, Science (80-. )., 348(6237), 895–899, doi:10.1126/science.aaa1668, 2015.*

*Beck, H. E., Vergopolan, N., Pan, M., Levizzani, V., van Dijk, A. I. J. M., Weedon, G. P., Brocca, L., Pappenberger, F., Huffman, G. J. and Wood, E. F.: Global-scale evaluation of 22 precipitation datasets using gauge observations and hydrological modeling, Hydrol. Earth Syst. Sci., 21(12), 6201–6217, doi:10.5194/hess-21-6201-2017, 2017.*

*Beck, H. E., Pan, M., Roy, T., Weedon, G. P., Pappenberger, F., Van Dijk, A. I. J. M., Huffman, G. J., Adler, R. F. and Wood, E. F.: Daily evaluation of 26 precipitation datasets using Stage-IV gauge-radar data for the CONUS, Hydrol. Earth Syst. Sci., 23(1), 207–224, doi:10.5194/hess-23-207-2019, 2019.*

*Boke-Olén, N.: Global Savannah Phenology, Lund University, Faculty of Science, Department of Physical Geography and Ecosystem Science, Lund., 2017.*

*Brandt, M., Rasmussen, K., Peñuelas, J., Tian, F., Schurgers, G., Verger, A., Mertz, O., Palmer, J. R. B. and Fensholt, R.: Human population growth offsets climate-driven increase in woody vegetation in sub-Saharan Africa, Nat. Ecol. Evol., 1(March), 0081, doi:10.1038/s41559-017-0081, 2017.*

*Brandt, M., Wigneron, J. P., Chave, J., Tagesson, T., Penuelas, J., Ciais, P., Rasmussen, K., Tian, F., Mbow, C., Al-Yaari, A., Rodriguez-Fernandez, N., Schurgers, G., Zhang, W., Chang, J., Kerr, Y., Verger, A., Tucker, C., Mialon, A., Rasmussen, L. V., Fan, L. and Fensholt, R.: Satellite passive microwaves reveal recent climate-induced carbon losses in African drylands, Nat. Ecol. Evol., 2(5), 827–835, doi:10.1038/s41559-018-0530-6, 2018.*

Fan, Y., Miguez-Macho, G., Jobbágy, E. G., Jackson, R. B. and Otero-Casal, C.: Hydrologic regulation of plant rooting depth, Proc. Natl. Acad. Sci. U. S. A., 114(40), 10572–10577, doi:10.1073/pnas.1712381114, 2017.

Kolus, H. R., Huntzinger, D. N., Schwalm, C. R., Fisher, J. B., McKay, N., Fang, Y., Michalak, A. M., Schaefer, K., Wei, Y., Poulter, B., Mao, J., Parazoo, N. C. and Shi, X.: Land carbon models underestimate the severity and duration of drought's impact on plant productivity, Sci. Rep., 9(1), 1–10, doi:10.1038/s41598-019-39373-1, 2019.

Longo, M., Knox, R. G., Medvigy, D. M., Levine, N. M., Dietze, M. C., Kim, Y., Swann, A. L. S., Zhang, K., Rollinson, C. R., Bras, R. L., Wofsy, S. C. and Moorcroft, P. R.: The biophysics, ecology, and biogeochemistry of functionally diverse, vertically and horizontally heterogeneous ecosystems: The Ecosystem Demography model, version 2.2-Part 1: Model description, Geosci. Model Dev., 12(10), 4309–4346, doi:10.5194/gmd-12-4309-2019, 2019.

Smith, B., Wärlind, D., Arneth, A., Hickler, T., Leadley, P., Siltberg, J. and Zaehle, S.: Implications of incorporating N cycling and N limitations on primary production in an individual-based dynamic vegetation model, Biogeosciences, 11(7), 2027–2054, doi:10.5194/bg-11-2027-2014, 2014.

Weedon, G. P., Balsamo, G., Bellouin, N., Gomes, S., Best, M. J. and Viterbo, P.: The WFDEI meteorological forcing data set: WATCH Forcing Data methodology applied to ERA-Interim reanalysis data, Water Resour. Res., 50(9), 7505–7514, doi:10.1002/2014WR015638, 2014.

Xu, X., Medvigy, D., Powers, J. S., Becknell, J. M. and Guan, K.: Diversity in plant hydraulic traits explains seasonal and inter-annual variations of vegetation dynamics in seasonally dry tropical forests, New Phytol., 212(1), 80–95, doi:10.1111/nph.14009, 2016.

---

## Author Response (AR1)

Dear Dr. Ito,

Thank you for studying through the open discussion and for your favourable response. Based on the referee reports and your feedback, we have now finalized our major revision of the manuscript.

The largest improvement in the revised manuscript was the addition of the model experiment results for all four Sahel sites. We implemented this by including summary figures, containing the amplitudes and legacies of the disturbance impacts for each site and each scenario. This highly facilitates the comparison of the responses over the different sites, and we elaborated on this accordingly in the text by rewriting several paragraphs. We did keep the time-series for the Dahra site in order to illustrate the typical response in function of time since the disturbance.

Further main improvements include an additional Taylor diagram to evaluate the model performance inside the rainy season, a map with an overview of all Sahel sites, and an additional diagram to illustrate how the vegetation response was analyzed. To avoid distraction of our central message, we removed a paragraph and a figure showing the cumulative carbon uptake on the PFT level, as this result did not contribute enough to justify its addition in the manuscript. Finally, we also resolved the remaining minor concerns of the referees throughout the text.

As we discussed in the manuscript, we agree that it would be highly valuable to perform similar experiments on vegetation models which incorporate a more advanced hydrological scheme, such as the ED2 model or an improved version of LPJ-GUESS. However, we are currently still optimizing the parameterization of the ED2 model, so including these results is unfortunately out of the scope for the current paper.

On the following pages you can find the revised manuscript, where all changes were highlighted. Hopefully these improvements suffice for a favourable review.

Thank you and best regards,

On behalf of all authors,

Wim Verbruggen

[revised manuscript text omitted]